# Self-emergent vortex flow of microtubule and kinesin in cell-sized droplets under water/water phase separation

Hiroki Sakuta [1,2,9,10], Naoki Nakatani[1], Takayuki Torisawa[3], Yutaka Sumino [4✉], Kanta Tsumoto [5], Kazuhiro Oiwa [6,7✉] & Kenichi Yoshikawa [1,8]

By facilitating a water/water phase separation (w/wPS), crowded biopolymers in cells form droplets that contribute to the spatial localization of biological components and their biochemical reactions. However, their influence on mechanical processes driven by protein motors has not been well studied. Here, we show that the w/wPS droplet spontaneously entraps kinesins as well as microtubules (MTs) and generates a micrometre-scale vortex flow inside the droplet. Active droplets with a size of 10–100 μm are generated through w/wPS of dextran and polyethylene glycol mixed with MTs, molecular-engineered chimeric four-headed kinesins and ATP after mechanical mixing. MTs and kinesin rapidly created contractile network accumulated at the interface of the droplet and gradually generated vortical flow, which can drive translational motion of a droplet. Our work reveals that the interface of w/wPS contributes not only to chemical processes but also produces mechanical motion by assembling species of protein motors in a functioning manner.

[1] Faculty of Life and Medical Sciences, Doshisha University, Kyotanabe, Kyoto 610-0394, Japan. [2] Organization for Research Initiatives and Development, Doshisha University, Kyotanabe, Kyoto 610-0394, Japan. [3] Cell Architecture Laboratory, Structural Biology Center, National Institute of Genetics, Mishima, Shizuoka 411-8540, Japan. [4] Department of Applied Physics, Faculty of Advanced Engineering, WaTUS and DCIS, Tokyo University of Science, Katsushika, Tokyo 125-8585, Japan. [5] Division of Chemistry for Materials, Graduate School of Engineering, Mie University, Tsu, Mie 514-8507, Japan. [6] Advanced ICT Research Institute, National Institute of Information and Communications Technology, Kobe, Hyogo 651-2492, Japan. [7] Department of Life Science, Graduate School of Science, University of Hyogo, Ako, Hyogo 678-1297, Japan. [8] Center for Integrative Medicine and Physics, Institute for Advanced Study, Kyoto University, Kyoto, Kyoto 606-8501, Japan. [9] Present address: Center for Complex Systems Biology, Universal Biology Institute, The University of Tokyo, Meguro, Tokyo 153-8902, Japan. [10] Present address: Graduate School of Arts and Sciences, The University of Tokyo, Meguro, Tokyo 153-8902, Japan. ✉email: ysumino@rs.tus.ac.jp; oiwa@nict.go.jp

Cells undertake a large number of different biochemical processes to live. These biochemical, and generally enzymatic processes need to be separated and confined to allow several reactions to proceed simultaneously and improve the reaction efficiency. Eukaryotic cells contain various membranous organelles, such as the nucleus, endoplasmic reticulum, lysosomes, and mitochondria. In addition to such membranous compartments, membraneless organelles[1–5] as well as coacervate[6,7], have recently been studied under the notion of liquid-liquid phase separation[8,9].

Cytoplasm, the contents within a cell surrounded by a cellular lipid membrane, is a crowded solution of biopolymers, i.e., proteins and nucleic acids, on the order of 30–50 wt%. Such a crowded polymer solution undergoes phase separation driven by an entropic depletion effect[10]; so-called water/water phase separation (w/wPS). In the present article, we adopted the term w/wPS to interpret the formation of aqueous microdroplets surrounded by aqueous solution, which is a cause of aqueous two-phase system (ATPS) often used in chemical engineering[9,11–13]. Aqueous solution of dextran (DEX) and polyethylene glycol (PEG) is a typical example of ATPS and w/wPS[14–18]. Both DEX and PEG are hydrophilic polymers that are readily soluble in water. However, their mixture induces w/wPS due to a reduced contribution of conformation entropy for polymer solutions (i.e., driven by depletion interaction). One of the characteristics of w/wPS is its ultralow interfacial tension[19] whose order ranges from 1 to 100 $\mu$N m$^{-1}$. This low interfacial tension allows the droplet coalescence to slow down and produce a long-lived micrometre scale droplet. Variation of partitioning ability of chemicals are another feature of w/wPS. Aqueous chemicals and biopolymers may be distributed homogeneously in both the phases, one of the phases, or at the interface of the w/wPS. These distributions can be controlled by pH, additional ions, and molecular size. Such characteristics are the foundation of the ATPS method based on w/wPS.

In terms of the artificial cell model system, w/wPS plays a vital role. Membranous systems have been proposed as potential cell model systems (e.g., liposomes (vesicles)[20–22] and water-in-oil emulsions[23,24]), which are coated by lipids and/or surfactants. However, w/wPS droplets have several advantages over liposomes and emulsions. In a droplet generated through w/wPS, biopolymers are expected to be partitioned spontaneously. This implies that no delicate preparation processes are required for w/wPS droplets compared to membranous compartments (i.e., vesicles and water-in-oil emulsions). Mechanical mixing of the chemical components provides a w/wPS droplet. Another advantage is the ease of supplying a chemical to w/wPS droplets. When charged chemicals need to be supplied into the droplet, the chemicals can be simply added to the external solution. For example, ATP can be supplied continuously to a droplet containing protein motors, thereby keeping the droplet active indefinitely. Such a simple scheme suggests that a w/wPS droplet may be a useful container for an artificial cellular environment[14–18]. Indeed, we previously showed the spontaneous localization of cytoskeleton, actin network, and DNA by adopting a type of w/wPS[17]. Here, short-strand DNA and G-actin were distributed in both PEG-rich/DEX-rich phases, while long-strand DNA and F-actin preferred DEX-rich phases. Furthermore, highly polymerized, long F-actin has been shown to localize at the interface between PEG-rich/DEX-rich phases[17]. This sensitivity of the partitioning to the molecular size is attributable to the different manner of macromolecule packing between the PEG-rich and DEX-rich phases caused by the depletion effect owe to the large difference of polymer flexibility[25] and unique characteristics of w/wPS droplet.

Despite of aforementioned roles of w/wPS in vivo and in vitro cell model systems, the mobility feature has been overlooked.

Spontaneous movement, one of the essential features of living systems, is achieved by consuming chemical energy, such as ATP, under isothermal conditions. On the contrary, most studies on droplets made through w/wPS have considered them as passive objects incapable of self-propelled motility, while membranous systems have incorporated motility successfully by adopting cytoskeletal and motor proteins[26–33]. In the present study, we included cytoskeleton proteins into a w/wPS droplet composed of a DEX-PEG mixture to activate and allow the droplet to propel itself in an organized manner. We focused on the stable formation of the vortical structure of flow inside a droplet and droplet motility. To this end, we created a DEX-PEG droplet system by including a mixture of microtubules (MTs) and a member of the kinesin family, 4-headed KIF5B$_{head}$-Eg5$_{tail}$. Notably, this mixture produces an aster-like structure often observed in the cluster of filaments and bridging motor proteins[34–36]. The presence of the aster-like structure leads the generation of active contractile stress similar to that observed in the actomyosin system[37,38].

## Results and discussion

### Formation of aster-like structure with MTs and chimeric kinesin construct, 4-headed KIF5B$_{head}$-Eg5$_{tail}$.

In this study, we used a mixture of MTs and a member of the kinesin family, 4-headed KIF5B$_{head}$-Eg5$_{tail}$, which locally produce active contractile stress, as in the actomyosin system[37,38]. As confirmed in our previous study[39], this combination of MTs and the kinesin rapidly forms aster-like structures within 5 min, when they are mixed under conditions appropriate for motility (Fig. S1a) in the absence of DEX and PEG. An aster-like structure in the current study is defined as a dynamic structure consisting of a radial array of MT with a node at its centre, where kinesin is concentrated. The plus-end of the MT points inward, in the direction of the node that is accumulated by the kinesin activity. Aster-like structures bridged by kinesin (i.e., aster-kinesin complexes) create a contractile network as schematically shown in Fig. S1b, a scheme that has been confirmed numerically. Such a mesoscale structure is in clear contrast to the well-known MT-kinesin system that generates extensile stress[40].

### Preparation of w/wPS droplet.

In this study, we used a PEG:DEX concentration = 5:5 (wt%), which is close to their critical points[17,41] and appropriate for producing small droplets via w/wPS. After mixing with a vortex shaker for a few seconds, droplets of DEX-rich phase were formed in a continuous PEG-rich phase. Under these conditions, droplets with diameters ranging between 10–100 $\mu$m keep their sizes for several hours due to their low interfacial tension. In the following experiment, we mainly observed droplets with a diameter of around 100 $\mu$m. Further, the appropriate amount of proteins, ATP, and ions was mixed with the aqueous PEG and DEX solution to obtain the preparation solution. Next, the preparation solution was mechanically mixed with a vortex shaker for a few seconds to create droplets. The obtained droplet took a semi-spherical shape having contact with a bottom slide glass as shown in Supplementary Fig. S2.

### Self-assembly of active droplet under w/wPS.

When MTs, DEX, PEG, and water in the absence of ATP is used as a preparation solution (see Supplementary Table S1 for the detailed concentration), MTs were spontaneously localized in DEX-rich droplets. Furthermore, the MTs spontaneously accumulated at the droplet interface by forming an artificial cortex (Fig. 1a). Actin filaments have been reported to form such a cortex due to the depletion interaction induced by their large persistence length, typically a few 10 $\mu$m[17,42]. Since MTs have even larger persistence lengths (0.1–10 mm)[43] than actin filaments, we

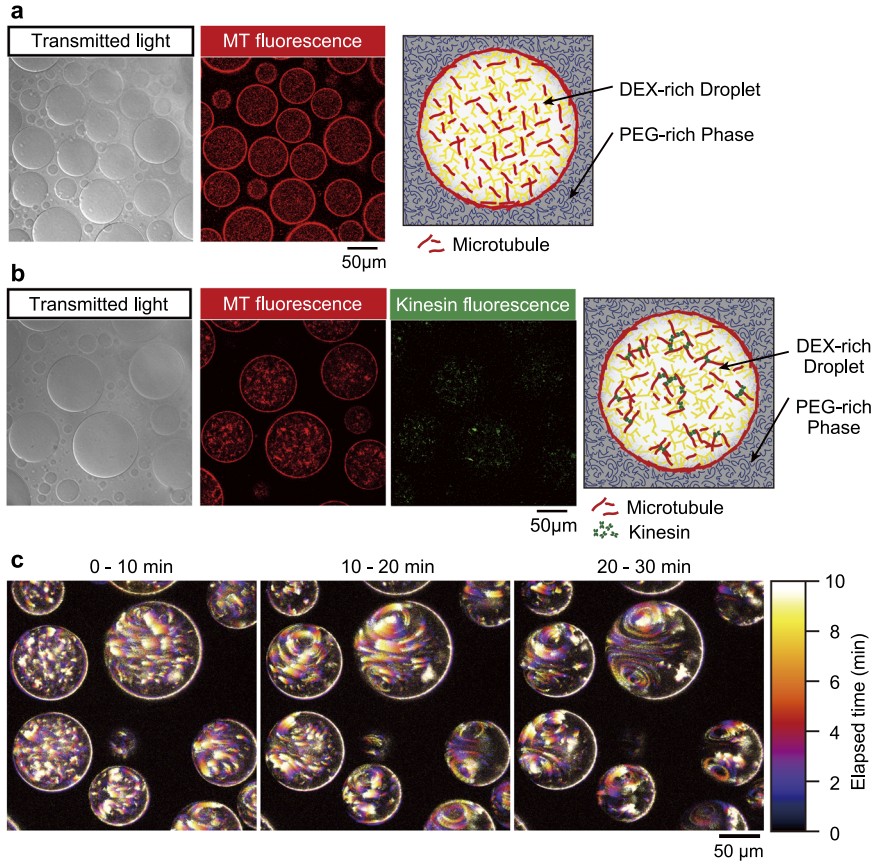

**Fig. 1 Spontaneous formation of vortex flow inside a w/wPS droplet, entrapping MTs and kinesins through self-construction from a PEG/DEX aqueous solution. a** Droplet generated from a PEG/DEX solution containing MTs. Mechanical mixing of a PEG/DEX binary solution close to the critical point and MTs spontaneously generates cell-sized DEX-rich droplets. MTs are distributed in the droplet and clearly visualized in fluorescent images. They are localized at the interface to make an artificial cortex. **b** Droplet generated with MTs and kinesins in the absence of ATP. Addition of kinesin tends to cross-link MTs. The accumulation of kinesin in the aggregates was confirmed by the fluorescent images. The cortical structure was preserved with the addition of kinesin. **c** The generated flow is driven by the chemical energy dissipation of ATP. Time development of the vortex flow inside of droplets. The snapshots are superimposed images for 10 min, where the flow was visualized based on the presence of MTs. Pseudo colour represents the elapsed time. A clear pair of vortices appear 20 min after mixing.

assume that depletion interaction also forms a cortex of MTs in the current system[44]. Upon the addition of kinesin without ATP, kinesin is distributed in the MT-rich region (see Supplementary Table S2 for detailed concentration). Kinesins crosslink MTs and form small aggregations in a droplet in addition to the cortex (Fig. 1b). Overall, in the absence of ATP, the formed cortex and aggregations were static, that is, neither flow nor motion was observed.

When ATP was included in a preparation solution (see Supplementary Table S3 for the detailed concentration), vortical flow was generated inside a droplet. Here, the ATP concentration used was 10 mM, a concentration 100 times higher than that observed when kinesin activity is saturated[45]. We mixed the preparation solution using vortex shaker and the vortex emerged after an induction period ranging from one to ten min (Fig. 1c, Supplementary Movie S1). This duration of the induction period correlates well with the time required for the formation of aster-like structure. We confirmed the formation of aster-like structure within a droplet, and the subsequent aggregate formation was confirmed via the bulk experiment with the presence of DEX-PEG (Fig. S1c, d). The MTs are found to gradually adsorb at the interface from a droplet phase confirmed via the density of fluorescent label (supplementary Fig. S1e). The observed growth of the size of MT/kinesin aggregates also supports that the created protein complex produces contractile stress.

Once the vortical flow appeared (Fig. 2a), the pair of vortices were stable for at least 60 min. Kymograph analysis of the interface (Fig. 2c) revealed that MTs accumulated to form a cap-like structure at the sink part accompanied by the interfacial flow. Based on Particle Image Velocimetry (PIV) analysis[46], the vortical flow inside of the droplet is visualized (Fig. 2d, schematically also depicted in Fig. 2b). We observed the flow from source to sink direction near the interface, and the strong backflow (sink to source direction) at the central part of the droplet. The speed of the flow was around 100 nm s$^{-1}$ (as shown in Supplementary Fig. S3). This speed is comparable to the typical speed of a motor observed in our previous study ($141 \pm 0.1$ nm s$^{-1}$)[39]. The vortical core gradually shifts in the direction of the source, indicating that the cap structure does not contribute to flow formation (Supplementary Movie S2). The velocity of the vortical flow shows positive correlation with the concentration of kinesin, $C_K$ (Supplementary Fig. S3). All of these observations are consistent with the notion that contractile stress produced by the aster-kinesin complex induces flow inside the droplet.

**W/wPS droplet as a mesoscale transporter**. We have shown that the assembly of active w/wPS droplet is spontaneous and robust with mechanical mixing. Our experiments further revealed the long lifetime of produced vortex flow in the droplet. The lifetime

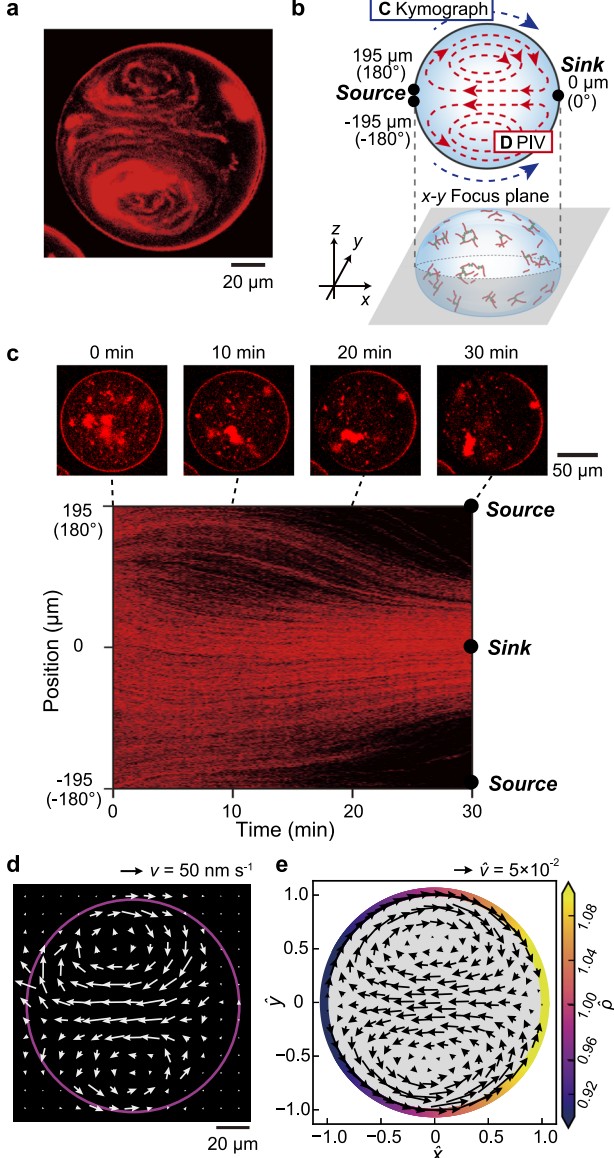

**Fig. 2 Analysis of active vortices inside a droplet upon the addition of ATP. a** MT labelled with ATTO647N is visualized at the middle part along the dorsoventral axis of the droplet. Addition of ATP to the system promotes the aggregation of MT, indicating the formation of an aster-like structure. Furthermore, active flow is generated at the interface, where MT and kinesin are accumulated to form an artificial cortex (Supplementary Movie S2). **b** Schematic illustration of droplet geometry and the coordinate system. **c**, Snapshots of the droplet every 10 min and kymograph analysis of the droplet interface. The fluorescently labelled MT and kinesin (aster-kinesin complex) gradually accumulated at the sink. **d** Flow field inside of a droplet obtained using the PIV method. While intensive flow is generated inside the droplet, accumulation of the artificial cortex was noted. The flow is directed source-to-sink at the interface, while back flow is noted at the middle. The double roll is clearly visualised and the flow speed is typically ~130 nm s$^{-1}$ (7800 nm min$^{-1}$), which is comparable to the speed of the accumulation of the aster-kinesin complex. **e** Numerical result of vortex flow profile at a $\hat{t} = 2000$ obtained using Eqs. (1–3) with experimentally plausible values. The colour indicates the MT surface density, while $\rho$ indicates the density gradient formation. Details of the numerical simulation as well as the estimation of parameters are given in the Method section.

of the convective motion may be extended by supplying kinesins, MTs and ATP to an active droplet from an external continuous phase. Thus, the present system has potential to be rechargeable actuator. Furthermore, the droplets were shown to exhibit a translation in the sink direction characterized by the cap-like structure of the aggregate (Fig. 3a, Supplementary Movie S3). The typical speed of the droplet motion was 42.1 ± 7.4 nm s$^{-1}$ ($N = 11$). This is comparable to the velocity of the vortex flow; however, an order of magnitude slower than the velocity of the kinesin on the filament[39] (Fig. 3b). The direction of the droplet translation in the field of view is different for each droplet and does not give apparent correlation. Conversely, the translation direction of an individual droplet correlates with positioning of the cap-like structure in the droplet. As the position of the cap-like structure indicates the sink direction of the vortex flow inside the droplet (Fig. 3c), the droplet translation should be driven by the vortex flow. These observations rule out the possibility of drift motion caused by external forces such as gravity. Translational motion of a droplet with a vortex flow inside has been previously reported[47–50]. Typically, the direction of translational motion of an active droplet in a fluid is opposite that in the present system. We infer that this opposite tendency is caused by the confined nature of the droplet and the effect of friction as well as its distorted shape. Nevertheless, a comprehensive analysis of the translational motion of the droplet is beyond the scope of this study. However, the observed translational motion of droplets confirms the possible application of w/wPS droplet for a spontaneously assembled mesoscale robot.

We note that the vortex directions within neighbouring droplets synchronize with each other (Fig. 3c). Such coupling of flow inside of droplets is probably possible by the concentration of ATP in the continuous phase and would be characteristic of w/wPS droplets where all phases are aqueous. Furthermore, the coupling may trigger coherent droplet motion that would be interesting to investigate especially in terms of collective behaviour of droplets.

The coupling between motion and deformation is also an interesting topic especially in terms of active matter physics[51]. W/wPS droplets have extremely low interfacial tension, of the order of $10^{-6}$ N m$^{-1}$, which is $10^{-4}$ of the typical interfacial tension value observed between ordinary fluids. Although a recently published study[52] showed intensive deformation of the interface, we could not identify noticeable deformation due to the flow within a droplet as shown in Figs. 1–3, and S2, where 3D imaging is depicted. This could be explained by the manner of force generation by the adsorbed aster-like structure at the interface. Such structure may create stress mainly tangential, not normal to the interface. Revealing the mechanism underlying such effect represents a fascinating topic for future study.

**Phase diagram and semi-quantitative mathematical model.** Controlling and understanding the vortex flow must be achieved to use and apply w/wPS droplet systems to molecular machinery. Thus, we examined the effect of the protein concentration on vortex formation. The phase diagram (Fig. 4a) revealed the preferential appearance of vortical flow for lower/higher concentrations of MT/kinesin, respectively. Figure 4b shows the abolition of the vortex upon the addition of MTs. When the concentration of MT was higher than 2.7 μM, neither the cap-like structure nor the vortical flow was observed (Supplementary Movie S4). These results indicate that the formation of the cap-like structure correlates well with vortex formation in the droplet.

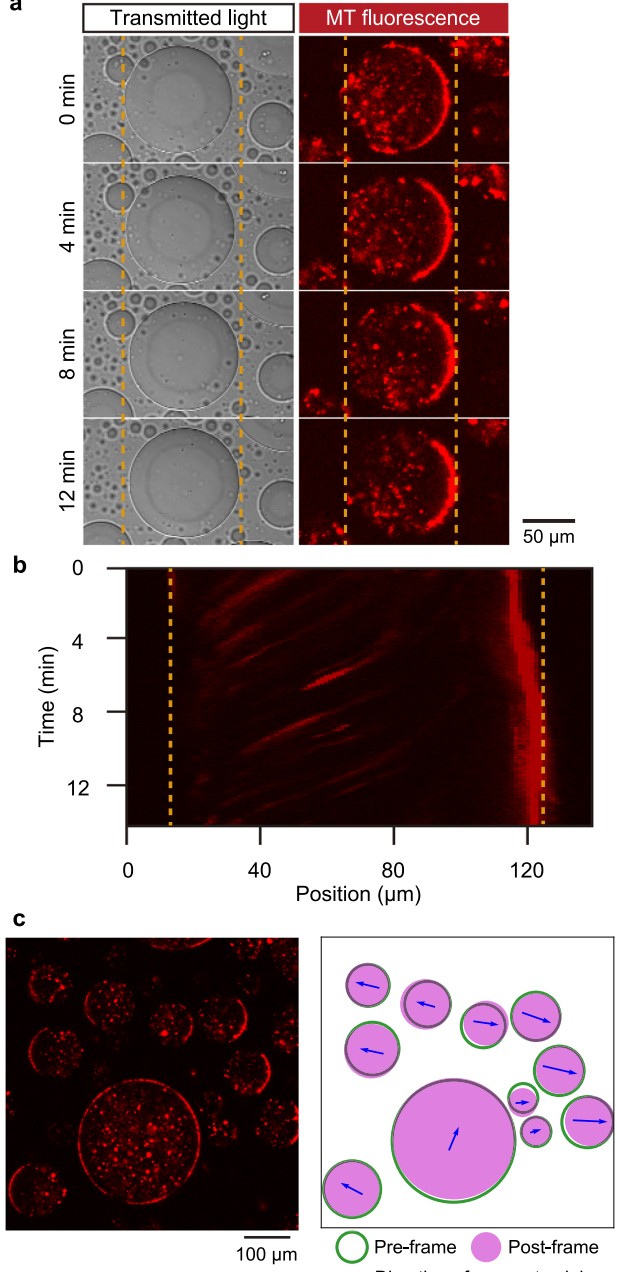

**Fig. 3 Translational motion of a droplet caused by vortical flow inside a droplet. a** Fluorescently labelled MTs are visualized in the fluorescent image while the droplet showed translational motion in the sink direction of a pair of vortices. **b** Kymograph taken at the middle of the droplet shows that the speed of the droplet was on the order of 10 nm s$^{-1}$. **c** The direction of the translational motion and the position of the cap-like structure at the sink part. Green coloured circle (edge) and magenta coloured circle (filled) indicate the position of droplets during the pre-frame and post-frame at interval 8.5 min. Blue arrow marks show the direction of the source-to-sink point, which are obtained via image analysis of the brightness-weighted centre of each droplet. Arrow size is correlated to the distance between droplet centre and the brightness-weighted centre.

The appearance of vortical motion with an increase in the kinesin concentration is attributed to the enhanced active contractile stress. The absence of a vortex with an increase in the MT concentration is due to the increase in hydrostatic pressure in the cortex. To illustrate this mechanism of droplet motion and the dependence of vortex formation on the protein concentration, here we discuss the mechanism of flow formation, and the phase diagram based on a simplified mathematical model that has been previously reported for active gel[53]. We assume that initially all MTs form aster-like structure, which produce active contractile stress with the addition of kinesin. To identify the essential features of self-organized vortical flow, we considered a 2-dimensional droplet for simplicity and assume Darcy's law at the interfacial layer, where the local velocity is proportional to the pressure gradient[54]. We focus on the initial regime of the flow structure in the droplet (corresponding initial 10 min). For simplicity, we neglect the viscosity in the cortex and assume the fluid motion is overdamped. In addition, a possible polarization and anisotropy of aster-like structure in the cortex is neglected; thus, the generated stress is assumed to be isotropic. The feedback from the flow and the concentration distribution in a droplet is also neglected (see 'Parameter estimation' within the Method section as well as the justification of this setup). A linear stability analysis based on the equation shown within the Method section revealed that the symmetric condition was broken when $C_K > \Xi + \Phi C_M$, where $C_M$ and $C_K$ are the bulk concentrations of MT and kinesin, and $\Xi$ and $\Phi$ are positive constants determined by the experimental conditions, respectively (Method section for details). In our model, the kinesin density $C_K$ contributes to the generation of active contractile stress that drives the spontaneous localization of cortex to have cap-like structure. Conversely, $C_M$ contributes to the hydrostatic pressure term that prevents localizing the cortex. Therefore, the obtained inequality shows that a larger $C_K$ and smaller $C_M$ are required for symmetry-breaking to cause vortical flow.

A numerical calculation based on this model reproduced the vortex formation inside of a droplet when coupled with a steady-state Stokes equation as shown in Fig. 2e and Supplementary Fig. S4, using FreeFem++ (Supplementary Movie S5)[55]. The calculation predicts that the profile of MT density at the surface breaks symmetry with its peak at the sink flow direction. This is consistent with the experimental observation where a cap-like structure was generated. The phase diagram for the protein concentration shown in Fig. 4a was successfully reproduced (Fig. 4c), where an increase in $C_K$ or a decrease in $C_M$ lead to vortex formation. The numerical simulations also reproduce the positive dependency of the flow velocity on $C_K$ (Supplementary Fig. S3). In the region with low $C_K$ and $C_M$, the prediction of our model deviates from the experimental results. This discrepancy could be due to the simplicity of our model, which overestimated the hydrostatic pressure at low $C_M$ and/or underestimated the contractile stress at low $C_K$. Nevertheless, the essential picture of droplet motion as well as semi-quantitative agreement with the experiment were established. Results of experiments that considered the droplet size as a parameter are shown in Supplementary Fig. S5.

We also identified several research questions that should be investigated in future studies. First, our experimental observation that the vortex core line gradually tilts to become parallel to the substrate with an increase in the kinesin concentration should be validated. Next, the relation between the directions of vortex flow and motion should be investigated in detail. Explaining these observations by extending the model for a 3-dimensional flow analysis[53,56] by adopting observed droplet shape (Supplementary Fig. S2) would add significant value to the existing scientific information.

## Conclusion

In this study, we demonstrated that mechanical mixing of an aqueous solution of MT and kinesin in an aqueous solution with PEG/DEX leads to the spontaneous formation of a droplet with

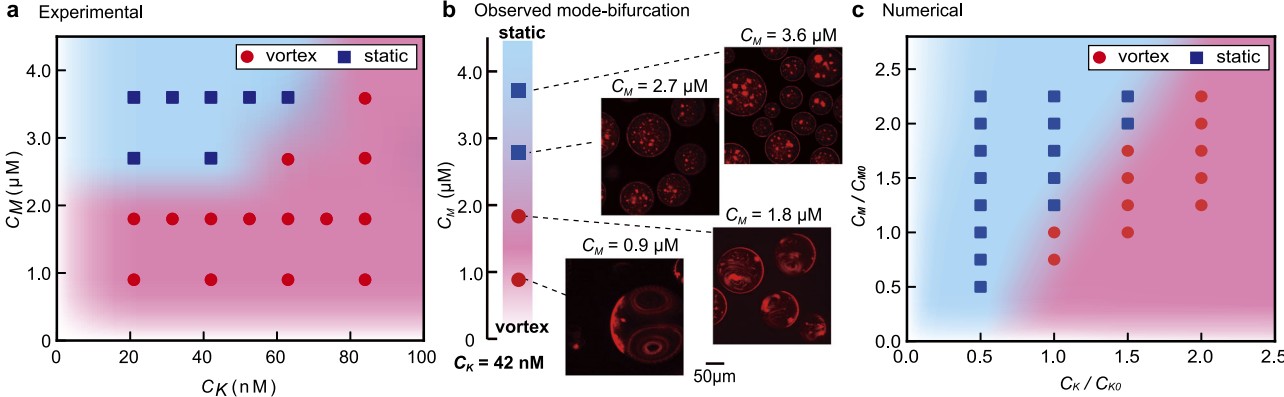

**Fig. 4 Behaviour of the phase-separated droplet in the presence of kinesin and MT. a** Phase diagram of droplet behaviour according to the kinesin and MT concentrations. Red circles and blue squares correspond to with or without vortical flow, respectively. In the upper-left corner of the diagram (low kinesin, high MT concentrations) vortical flow was absent and the aggregate formation inside the droplet was noted. **b** Superimposed fluorescent images captured every 10 min of droplet motion, at different concentrations of $C_M$. $C_M$ ceases the vortical motion. **c** Phase diagram based on numerical simulation under the assumption described in the main text. $C_K/C_{K0}, C_M/C_{M0} = 1$ corresponds to the parameters used to simulate the flow profile shown in Fig. 2e.

an artificial cortex. The droplet is composed of a DEX-rich phase in a PEG-rich aqueous phase, while MT and kinesin are distributed in droplets. A cortex made of MT and kinesin is formed spontaneously at the droplet interface and becomes active with the addition of ATP, which allows stable vortical flow for 1 h. The combination of MT and kinesin used in the current study facilitates the formation of an aster-like structure after 5 min and provides an effective contractile network subsequently. Notably, the current artificial cortex is generated spontaneously using the interface of a droplet as a mould. Furthermore, the generated droplets cause the vortical flow inside droplets, which also produces a flow outside the droplets.

Recent progress in molecular nano-/bio-technology led to the creation of various molecular machines including molecular motors at a nanometre scale. Assembly of these molecular machines often involves delicate and complicated procedures that are only possible through human intervention. In the present system, the vortical flow is formed using the interface of w/wPS as a template for motor proteins to self-organize into an appropriate cortex geometry. Our work reveals that the interface of w/wPS can be used for assembling various molecular machines that can function in an organized manner. Their convenience as well as possible use as a supply route of water-soluble chemicals to a droplet give w/wPS droplets advantage over vesicles and emulsions that contain a membrane structure. The application of w/wPS with molecular nano-machinery is expected to facilitate the formation of self-organized micrometre scale actuators and robots.

## Methods

**Polymers**. To generate a water/water microdroplet, we used PEG and DEX. PEG 6,000 (molecular weight (Mw) = 7300–9300 Da) and DEX (Mw = 180,000–210,000 Da) were purchased from FujiFilm Wako Pure Chemical Industries (Osaka, Japan) and stocked at a concentration of 20 wt% in nuclease-free water (Milli-Q, 18.2 MΩ cm).

**Kinesin**. We used the 4-headed kinesin, Eg5, consisting of two motor units which have two heads at each pole. Eg5 can cross-link MTs. In this study, we generated a chimeric Eg5 which has a head consisting of KIF5 (KIF5B_head-Eg5_tail)[39]. Kinesin was fused to a green fluorescent protein (eGFP, Ex: 488 nm, Em: 509 nm) for fluorescence microscopy.

**Microtubule**. MTs were obtained through the polymerization of tubulin as previously described[39]. To perform fluorescence microscopy, we fluorescently-labelled tubulin with ATTO647N (Ex: 646 nm, Em: 664 nm, ATTO-TEC GmbH, Siegen, Germany) which contained 6% of total tubulin. MT was polymerized with Guanosine 5'-Triphosphate (GTP) and stabilized with 40 μM paclitaxel (Taxol, Sigma-Aldrich, St. Louis, MO, USA).

**Water/water microdroplets**. In this study, we used a PEG:DEX concentration = 5:5 (wt%), which is in the vicinity of a binodal line[17,41]. Microdroplets with diameters ranging between 10 to 100 μm could remain the same size for several hours after being generated using mechanical mixing with a vortex shaker. The microdroplets consisted of a DEX-rich internal water phase surrounded by a PEG-rich external water phase. The localization of MT or MT and kinesin was confirmed using solutions with compositions listed in Supplementary Tables S1 and S2. All experiments were performed at room temperature. The composition of the experimental solution allowing the formation of vortical flow inside of droplets with ATP and MgSO4 is listed in Supplementary Table S3.

**Microscopy**. Images were obtained using confocal-laser scanning microscopy (Nikon A1, Nikon, Tokyo, Japan). The obtained images were analysed using ImageJ software[57] and MATLAB (MathWorks, Natick, MA, USA) with a plugin for Particle Image Velocimetry (PIV), PIVlab[46].

**Numerical model**. In our mathematical model, we considered the system's basic elements MT as an aster-like structure, which can flow and migrate in a droplet. Our microscopic observation made it clear that such MT can attach at the droplet interface and form a thick cortex while producing flow. MT with an aster-like structure can produce contractile stress when a kinesin motor is present. We built the following dynamical equation based on a previous study that investigated the contractile cortex[53] while considering a droplet as a 2-dimensional cylinder with diameter R. Notably, the motors and aster-like structures in a single droplet gradually decreased their number after approximately one hour by creating the accumulated cortex at the interface. We focused on the initial regime of the flow structure in the droplet (corresponding initial 10 min). In such an initial regime, we assumed that the available concentration of motors and aster-like structures in a droplet is fixed and constant. For simplicity, we ignored the viscosity within the cortex. Furthermore, the possible polarization and anisotropy of an aster-like structure in the cortex was neglected. Therefore, we assumed that the generated stress is isotropic. Here, $\rho$ and $\mu$ represent the local densities of MT and kinesin at the interface. A kinesin motor produces active contractile stress $-\zeta\mu$, where $\zeta < 0$. We considered that the cortex follows the Darcy's law[54]. MT and kinesin show adsorption and desorption from the bulk, and the concentration in the droplet is assumed to be homogeneous (i.e., the effect of flow in the bulk is neglected). The dynamics are governed by the following equations:

$$\frac{\partial \rho}{\partial t} + \frac{1}{R}\frac{\partial}{\partial \theta}(\rho v) = -(k\rho - k_+ C_M), \tag{1}$$

$$\frac{\partial \mu}{\partial t} + \frac{1}{R}\frac{\partial}{\partial \theta}(\mu v) = -(\kappa\mu - \kappa_+ C_K) + \frac{D}{R^2}\frac{\partial^2 \mu}{\partial \theta^2}, \tag{2}$$

and

$$v = \frac{1}{\xi R}\frac{\partial}{\partial \theta}\left(-\zeta\frac{\mu}{\mu_0} - \alpha\frac{\rho}{\rho_0} + \frac{\beta}{R^2}\frac{\partial^2}{\partial \theta^2}\frac{\rho}{\rho_0}\right). \tag{3}$$

where, $k_+$ ($k$) and $\kappa_+$ ($\kappa$) are the adsorption (desorption) rate of MT and kinesin to (from) the interface, and $C_M$ and $C_K$ are actual control parameters (i.e., bulk concentrations of MT and kinesin, respectively). Equations (1) and (2) describe the adsorption equilibrium of MT (kinesin) between bulk and the interface. The diffusion of MT is neglected while those of kinesin is introduced in Eq. (2); thus, reflecting the large difference in molecular size associated with the inclusion of the diffusion term. Equation (3) describes the active contraction by the first term, the

hydrostatic pressure of MT by the second term, where the third term induces the aggregating behaviour of MT. Further, $\rho_0$ and $\mu_0$ represent the reference density of MTs and kinesins (see Parameter estimation). The hydrostatic pressure is induced by the surface density of MT, and a gradient term is introduced so that MT can have a homogeneous density. For compact notation, we introduce units such that $\rho = \frac{k_+ C_M}{k}\hat{\rho}$, $\mu = \frac{\kappa_+ C_K}{k}\hat{\mu}$, $t = \hat{t}/k$, and $v = kR\hat{v}$, where time and space are normalized by $1/k$ and $R$. We obtain the following dimensionless expressions of dynamical equations:

$$\frac{\partial \hat{\rho}}{\partial \hat{t}} + \frac{\partial}{\partial \theta}(\hat{\rho}\hat{v}) = -(\hat{\rho} - 1), \tag{4}$$

$$\frac{\partial \hat{\mu}}{\partial \hat{t}} + \frac{\partial}{\partial \theta}(\hat{\mu}\hat{v}) = -\hat{\kappa}(\hat{\mu} - 1) + \hat{D}\frac{\partial^2 \hat{\mu}}{\partial \theta^2}, \tag{5}$$

and

$$\hat{v} = \bar{\zeta}\frac{\partial \hat{\mu}}{\partial \theta} - \hat{\alpha}\frac{\partial \hat{\rho}}{\partial \theta} + \hat{\beta}\frac{\partial^3 \hat{\rho}}{\partial \theta^3}, \tag{6}$$

Where $\hat{\kappa} = \kappa/k$, $\hat{D} = \frac{D}{kR^2}$, $\bar{\zeta} = -\hat{\zeta} = -\frac{\zeta}{k\xi R^2}\frac{\kappa_+ C_K/k}{\mu_0}$, $\hat{\alpha} = \frac{\alpha}{k\xi R^2}\frac{k_+ C_M/k}{\rho_0}$, and $\hat{\beta} = \frac{\beta}{k\xi R^4}\frac{k_+ C_M/k}{\rho_0}$. Note that $\hat{\zeta} < 0$ and $\bar{\zeta} > 0$, indicate active contraction.

A numerical simulation was conducted with $\hat{\kappa} = 1$, $\hat{D} = 0.001$, $\hat{\alpha} = 1$, $\hat{\beta} = 1$, and $\bar{\zeta} = 3.1$, which correspond to the conditions associated with Fig. 2e and Supplementary Fig. S4a. The flow inside a droplet was assessed based on the assumption that the flow obeys Stokes equation and satisfies a steady state determined by the cortex flow as a boundary condition at the droplet interface. To obtain the results depicted in Fig. 3c, we varied the parameters $\hat{\alpha}$, $\hat{\beta}$, and $\bar{\zeta}$ in a relative manner, according to the dependence $\hat{\alpha} \propto C_M$, $\hat{\beta} \propto C_M$, and $\bar{\zeta} \propto C_K$. Normalized $C_K/C_{K0}$ and $C_M/C_{M0}$ in Fig. 4c are $\bar{\zeta} = 3.1$ and $\hat{\alpha} = \hat{\beta} = 1$; i.e., the state $(1, 1)$ corresponds to conditions shown in Fig. 2e and Supplementary Fig. S4a (Supplementary Movie S5). Explicit notations of $C_{K0}$ and $C_{M0}$ are given in Parameter estimation.

The phase diagram of Supplementary Fig. S5 was produced based on the assumption that the parameters: $\hat{\kappa} = 1$, $\hat{D} = 0.001$, $\hat{\alpha} = 1$, $\hat{\beta} = 1$, and $\bar{\zeta} = 3.1$, which corresponds to $R = 10^{-4}$ m, as the condition for $C_{K0}$ and $C_{M0}$. The parameters for different $R$ values were estimated by the dependence of $\hat{D} \sim R^{-2}$, $\bar{\zeta} \sim R^{-2}$, $\hat{\alpha} \sim R^{-2}$, and $\hat{\beta} \sim R^{-4}$. See also Parameter estimation for more details.

As mentioned above, the homogeneous solution was not stable for certain conditions and generated vortical flow as in the experiment. These criteria can be calculated with a simple linear stability analysis. A spatially homogeneous, steady-state solution is $\hat{\rho} = 1$ and $\hat{\mu} = 1$. Next, we considered fluctuation around the steady-state value, $\delta\hat{\rho} = \hat{\rho} - 1$ and $\delta\hat{\mu} = \hat{\mu} - 1$, and obtained the following equations:

$$\frac{\partial \delta\hat{\rho}}{\partial \hat{t}} = -\delta\hat{\rho} - \left[\bar{\zeta}\frac{\partial^2 \delta\hat{\mu}}{\partial \theta^2} - \hat{\alpha}\frac{\partial^2 \delta\hat{\rho}}{\partial \theta^2} + \hat{\beta}\frac{\partial^4 \delta\hat{\rho}}{\partial \theta^4}\right] \tag{7}$$

$$\frac{\partial \delta\hat{\mu}}{\partial \hat{t}} = -\hat{\kappa}\delta\hat{\mu} + \hat{D}\frac{\partial^2 \delta\hat{\mu}}{\partial \theta^2} - \left[\bar{\zeta}\frac{\partial^2 \delta\hat{\mu}}{\partial \theta^2} - \hat{\alpha}\frac{\partial^2 \delta\hat{\rho}}{\partial \theta^2} + \hat{\beta}\frac{\partial^4 \delta\hat{\rho}}{\partial \theta^4}\right] \tag{8}$$

We can set $\delta\hat{\rho} = \sum_{q=-\infty}^{\infty} \hat{\rho}_q e^{iq\theta}$ and $\delta\hat{\mu} = \sum_{q=-\infty}^{\infty} \hat{\mu}_q e^{iq\theta}$, leading to the expression:

$$\frac{d}{d\hat{t}}\begin{pmatrix} \hat{\rho}_q \\ \hat{\mu}_q \end{pmatrix} = \begin{pmatrix} -1 - (\hat{\alpha} + \hat{\beta}q^2)q^2 & \bar{\zeta}q^2 \\ -(\hat{\alpha} + \hat{\beta}q^2)q^2 & -\hat{\kappa} + (\bar{\zeta} - \hat{D})q^2 \end{pmatrix}\begin{pmatrix} \hat{\rho}_q \\ \hat{\mu}_q \end{pmatrix} = A\begin{pmatrix} \hat{\rho}_q \\ \hat{\mu}_q \end{pmatrix} \tag{9}$$

with characteristic equation:

$$\det\begin{pmatrix} -1 - (\hat{\alpha} + \hat{\beta}q^2)q^2 - \lambda_q & \bar{\zeta}q^2 \\ -(\hat{\alpha} + \hat{\beta}q^2)q^2 & -\hat{\kappa} + (\bar{\zeta} - \hat{D})q^2 - \lambda_q \end{pmatrix} = 0 \tag{10}$$

$$\lambda_q^2 - (\text{tr}A)\lambda_q + \det A = 0 \tag{11}$$

The linear stability is broken by $\text{Re}[\lambda_q] > 0$ when either $\text{tr } A > 0$ or $\det A < 0$. The $\text{tr } A > 0$ is realized when:

$$\bar{\zeta}q^2 > \hat{\kappa} + \hat{D}q^2 + 1 + (\hat{\alpha} + \hat{\beta}q^2)q^2 = f(q), \tag{12}$$

and a $\det A < 0$ is realized when:

$$\bar{\zeta}q^2 > (\hat{\kappa} + \hat{D}q^2)\{1 + (\hat{\alpha} + \hat{\beta}q^2)q^2\} = g(q). \tag{13}$$

Neither $\text{tr } A > 0$ nor $\det A < 0$ is satisfied when $q = 0$, indicating that homogeneous perturbation is always stable.

We observe that:

$$g(q) - f(q) = (\hat{\kappa} + \hat{D}q^2)\{1 + (\hat{\alpha} + \hat{\beta}q^2)q^2\} - (\hat{\kappa} + \hat{D}q^2 + 1 + (\hat{\alpha} + \hat{\beta}q^2)q^2)$$
$$= \hat{\kappa} + \hat{D}q^2 + (\hat{\kappa} + \hat{D}q^2)(\hat{\alpha} + \hat{\beta}q^2)q^2 - (\hat{\kappa} + \hat{D}q^2 + 1 + (\hat{\alpha} + \hat{\beta}q^2)q^2)$$
$$= (\hat{\kappa} + \hat{D}q^2 - 1)(\hat{\alpha} + \hat{\beta}q^2)q^2 - 1 \tag{14}$$

When a pair of vortices appears, the stability of $q = 1$ is broken. From the above expression, we also observe that:

$$g(1) - f(1) = (\hat{\kappa} + \hat{D} - 1)(\hat{\alpha} + \hat{\beta}) - 1. \tag{15}$$

With parameters $\hat{\kappa} = 1$, $\hat{D} = 0.001$, and $\hat{\alpha}, \hat{\beta} \sim 1$ which correspond to the conditions in our experiments, $g(1) < f(1)$. Thus, instability occurred for $q = 1$, with the condition:

$$\bar{\zeta} > (\hat{\kappa} + \hat{D})(1 + \hat{\alpha} + \hat{\beta}) \tag{16}$$

Indeed, we confirmed tr $A < 0$ for the condition $\hat{\kappa} = 1$, $\hat{D} = 0.001$, and $\hat{\alpha}, \hat{\beta} = 1$, with $\bar{\zeta} = 3.1$. This indicates there occurs the saddle-node bifurcation.

The condition for the bifurcation can be Eq. (16) is rephrased as:

$$C_K > \frac{\mu_0(\kappa R^2 + D)}{|\zeta|k\kappa_+\rho_0 R^4}\left\{k^2\xi R^4\rho_0 + k_+ C_M(R^2\alpha + \beta)\right\} \tag{17}$$

The parameters in the main text can be described as:

$$\Xi = \frac{\mu_0(\kappa R^2 + D)k\xi}{|\zeta|\kappa_+}, \quad \Phi = \frac{\mu_0(\kappa R^2 + D)k_+}{|\zeta|k\kappa_+\rho_0 R^4}(R^2\alpha + \beta) \tag{18}$$

and

$$C_K > \Xi + \Phi C_M \tag{19}$$

Thus, larger $C_K$ and smaller $C_M$ are required for symmetry-breaking, which is consistent with our experimentally observed phase diagram.

Further calculation leads that the observed bifurcation is supercritical pitch-fork bifurcation. Detailed calculation to find the reduced expression with the ones close to currently used parameters, $\hat{\kappa} = 1$, $\hat{D} = 0$, $\hat{\alpha} = 1$, and $\hat{\beta} = 1$, is given in the supplementary text.

**Parameter estimation.** As parameters, we used $R = 10^{-4}$ m, and put it as $R_0$. At a reference density $\rho_0$ and $\mu_0$, we assume $k = 10^{-1}$ s$^{-1}$, which is comparable to $\kappa = 10^{-1}$ s$^{-1}$, the desorption rate of myosin from the cortex[58]. Next, $\xi = 10^{17}$ Pa s m$^{-2}$ was estimated from the friction coefficient of an actin network[59]. $R$, $k$, and $\xi$ are dimensionless expressions for space, time, and pressure. For $\alpha$, which is the inverse of the adiabatic compressibility of aster-like structures, we estimated its order from those of F-actin[60], $6.9 \times 10^{-10}$ Pa$^{-1}$, and used $\alpha = 10^8$ Pa, which is slightly smaller, and hence softer, for considering the sparce aster-like structure in the present study. We also used $D = 10^{-12}$ m$^2$ s$^{-1}$[59,61,62]. We obtain $\hat{\kappa} = 1$, $\hat{D} = 0.001$, and $\hat{\alpha} = 1$. At MT and kinesin concentration in a droplet $C_{M0} = 1.8$ μM and at $C_{K0} = 42$ nM, we assumed adsorption equilibrium density $k_+ C_{M0}/k$, coincide with reference density $\rho_0$ for the standard condition with active vortical flow. This implies $C_{M0} = \frac{k\xi R_0^2}{\alpha}\frac{k\rho_0}{k_+}$. Note that $\frac{k\xi R_0^2}{\alpha}$ is dimensionless and order 1 in our parameter estimation.

The remaining parameters were $\beta$ and $\zeta$. We determined corresponding dimensionless parameters $\hat{\beta}$ and $\bar{\zeta}$ to reproduce the vortical flows in numerical simulations. We noted the vortical flow was $v = 10^{-7}$ m s$^{-1}$, which corresponds to $\hat{v} = 10^{-2}$. To reproduce these values, we chose $\hat{\beta} = 1$ and $\bar{\zeta} = 3.1$, which are slightly above the threshold for instability estimated from the linear stability analysis. Taking these values as reference state implies: $C_{K0} = 3.1 \times \frac{k^2\xi R_0^2}{\kappa_+|\zeta|}\mu_0$, and $\beta = \alpha R_0^2$.

The obtained phase diagram with respect to size $R$ and $C_K$, is illustrated in Supplementary Fig. S5, where the given number is the number of vortex pairs. The parameter at $R = 10^{-4}$ m was set as a standard and the parameters for different $R$ values were estimated by the dependence of $\hat{D} \sim R^{-2}$, $\bar{\zeta} \sim R^{-2}$, $\hat{\alpha} \sim R^{-2}$, and $\hat{\beta} \sim R^{-4}$. Explicitly we may rewrite $\bar{\zeta} = 3.1 \times \frac{C_K}{C_{K0}}\left(\frac{R_0}{R}\right)^2$, $\hat{\alpha} = \frac{C_M}{C_{M0}}\left(\frac{R_0}{R}\right)^2$, and $\hat{\beta} = \frac{C_M}{C_{M0}}\left(\frac{R_0}{R}\right)^4$, where we used the explicit notation of $C_{M0}$, $C_{K0}$, and $\beta = \alpha R_0^2$.

To reproduce the bulk flow inside of the droplet, we used the numerical results of flow speed at the surface of the droplet as the boundary condition, and steady-state flow of a 2-dimensional Stokes equation was calculated as follows:

$$\hat{\eta}\hat{\nabla}^2\hat{v} = \hat{\nabla}\delta\hat{p} \tag{20}$$

using FreeFem++. The estimated Reynolds number was $10^{-9}$, which justifies the use of the Stokes approximation. The inertial terms in the time-dependent Stokes equation were considered negligible from:

$$\frac{\rho_w kR^2}{\eta}\frac{\partial \hat{v}}{\partial \hat{t}} = \hat{\nabla}^2\hat{v} - \frac{\xi R^2}{\eta}\hat{\nabla}\delta\hat{p} \tag{21}$$

where $\frac{\rho_w kR^2}{\eta} = 10^{-5}$. To estimate the fluctuation under pressure, $\delta\hat{p}$, we used $\hat{\eta} = \frac{\eta}{\xi R^2} = 10^{-10}$.

For the phase diagram illustrated in Fig. 4c for different $C_K$ and $C_M$ (hence, $\hat{\alpha}, \hat{\beta}, \bar{\zeta}$) were varied from $\hat{\kappa} = 1$, $\hat{D} = 0.001$, $\hat{\alpha} = 1$, $\hat{\beta} = 1$, and $\bar{\zeta} = 3.1$, which corresponds to $C_{K0}$ and $C_{M0}$.

**Numerical simulation.** Numerical integration was conducted based on an explicit scheme, with the discretization $\Delta t = 10^{-9}$ and $\Delta\theta = 2\pi/256$ under a periodic boundary condition. The simulation starts at $\hat{t} = 0$, from the random initial condition adding homogeneous noise with a width $[-0.05:0.05]$ to the steady-state value of $\hat{\rho} = 1$, and $\hat{\mu} = 1$. The flow inside the droplet was simulated using FreeFem++ based on the following procedure. The one dimensional velocity fields $\hat{v}$ calculated with $\hat{\rho}$ and $\hat{\mu}$ was used as the boundary condition of tangential velocity at the interface of circular domain that represents the internal phase of a droplet. At each time step, the steady-state solution for the Stokes equation was calculated from $\hat{\eta}\hat{\nabla}^2\hat{v} = \hat{\nabla}\delta\hat{p}$. P2 mesh was used for $\hat{v}$, and P1 mesh was used for $\delta\hat{p}$. We used a penalty term $\epsilon\delta\hat{p}$, where $\epsilon = 10^{-6}$.

**Reporting summary.** Further information on research design is available in the Nature Portfolio Reporting Summary linked to this article.

## Data availability

Data supporting the findings of this manuscript and the kinesin constructs used are available from the corresponding authors upon reasonable request.

## Code availability

The custom computer code used to analyse the images and conduct the numerical simulation is available on Zenodo at https://doi.org/10.5281/zenodo.7784296 under CC BY 4.0 licence.

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

## Acknowledgements
This work was supported by JSPS KAKENHI Grant Numbers JP21K15057 to H.S., JP19H05403, JP21H00409, JP21H01004 to Y.S., JP17K07376 and JP21H02455 to K.O., and JP20H01877 to K.Y.

## Author contributions
H.S., N.N., T.T., K.O., and K.Y. conceived the project. H.S., N.N., and T.T. prepared the samples. H.S., N.N., T.T., and K.T. performed the measurements. H.S., N.N. and Y.S. built a mathematical model, while Y.S. performed the simulations. H.S. N.N. and Y.S. performed the data analysis and wrote the manuscript with feedback from their co-authors. K.O. and K.Y. supervised the project.

## Competing interests
The authors declare no competing interests.
