## [Peer Review file · Communications Chemistry]

Self-emergent vortex flow of microtubule and kinesin in cell-sized droplets under water/water phase separationReviewers' comments:

Reviewer #1 (Remarks to the Author):

In the manuscript entitled "Self-Emergent Vortex Flow of Microtubule/Kinesin in Cell-Sized Droplets under Water/Water Phase Separation," the authors describe a study in which they merge microtubules and molecular motors with a phase separating two-phase PEG/Dextran system.

The authors describe interesting experimental observations in which the cytoskeletal proteins strongly partition to the Dextran-rich phase. When confined within such a droplet the cytoskeletal components generate autonomous flows. In principle, these are interesting observations that could have an impact on the field of active matter. With that being said I do believe that some of the observations need to be described/quantified in more detail before the manuscript is acceptable for publication.

1. The authors claim that microtubules form an aster-like structure when confined within Dextran droplets. Asters have an outwardly splayed structure with motor proteins concentrated at their center. The authors need to demonstrate the presence of such structures, as has been done previously in uniform one-component systems. This could be accomplished by using high numerical aperture water immersion objectives as has been done for numerous other cytoskeletal structures.

2. The authors claim that vortex flows can translate droplets. This is an intriguing observation, albeit one that has been observed in several other cytoskeletal systems at significantly larger speeds. However, one needs to demonstrate a stronger connection between cytoskeletal structure, the flows they cause, and the resulting droplet motion. In particular, it would be helpful to simultaneously study the activity-powered motion of many droplets in one field of view to demonstrate that their directions are uncorrelated to each other. The motility of droplets is sufficiently slow that one needs to convincingly eliminate other possible causes of drift which could be external flows or even spatial changes in droplet confinement when the droplet diameter is smaller than the chamber thickness.

Reviewer #2 (Remarks to the Author):

The authors Sakuta et al. show that droplets spontaneously formed by depletant-induced phase separation can be used to enclose cytoskeletal gels (microtubule-kinesin mixtures). Interestingly, the motor activity leads to active behavior of the droplet leading to vortical flows within the droplet and their subsequent translation. Based on the observed flows of microtubule in the droplet, the authors provide a minimal mechanical model based on active gel theory that captures how droplet activity depends on a competition of motor and microtubule concentration.

There is considerable interest in reconstituted systems that exhibit life-like properties such as the ability to spontaneously move, in materials science and synthetic biology. As such, this work is a welcome addition to the considerable literature on strategies to create self-propelling droplets. The ability to enclose cytoskeletal gels in a simple type of water/water phase-separated droplet is certainly exciting.

However, there are important questions that the present study does not explore or answer (detailed below), which I think are imperative for a complete scientific study. These need to be addressed before the manuscript can be considered for publication.

1. Aster formation is not directly demonstrated/explained. The authors say they expect asters based on their previous work, "As confirmed in a previous study³², this combination of MTs and the kinesin rapidly forms aster structures". However, it is not explained what exactly these asters correspond to in the present experiment. Are these the bright clusters of MTs and kinesin seen in the bulk of the

droplet? Or do they mean the aggregation of MTs in the cortex, which is really what they use in their computational model?

- a. How do we know these asters are contractile based on the evidence presented in this paper?
- b. Can the authors visualize the radial alignment of the microtubules in these asters?
- c. What sets this MT-kinesin system apart from the active nematic system which gives extensile as opposed to contractile flows?

2. Reason for microtubule aggregation in droplets. The authors write "Depending on their chemical structure, biopolymers spontaneously accumulate in either the droplet's internal DEX-rich phase or its external PEG-rich phase". This requires further explanation. What about the "chemical structure" of microtubules determines that it associates with the DEX-rich droplet? (Is it size, charge, or something else?) Can microtubules also associate with PEG under other experimental conditions?

3. No exploration at all of droplet self-propulsion I am missing an explanation in the manuscript of how the droplet translational velocity depends on motor activity and why it moves in the way it does. The authors write, quite cryptically, "Typically, the direction of translational motion of an active droplet in a fluid is opposite that in the present system" and a "a comprehensive analysis of the translational motion of the droplet is beyond the scope of this study". While this maybe so, it seems unfair to the reader to leave them hanging without any explanation whatsoever. The authors should at the very least :

a. experimentally quantify how droplet propulsion velocity depends on motor activity (e.g. by varying ATP) or concentration. Discuss what are the important control parameters for droplet velocity.

b. extend the simple model to provide a mathematical relationship between droplet velocity and the important control parameters. This has already been done in ref. 40 by mapping the active cortical flows to Marangoni flows based on an effective surface tension gradient. This predicts a linear relationship between velocity and motor activity.

c. discuss qualitatively the reason for the droplet movement in a certain direction

4. Discrepancy between model and experimental phase diagrams. There is an obvious difference between experimental observation and the model prediction (Figs. 4a and 4c). The theoretical prediction is a linear relation between C_K and C_M , but the experiment deviates from that at low C_M . The authors have not commented at all on this, but it is very important that they at least discuss this and the possible drawbacks in their model.

5. Statement of assumptions in the model. There are several unstated assumptions in the simple model that the authors use, for example:

- a. viscosity within the cortical layer is ignored, and the fluid is treated as overdamped.
- b. possible polarization and anisotropy of the microtubule distribution and active stresses are ignored.
- c. The origin of the hydrostatic pressure is actually the resistance of the gel to deformation , so the parameter α is in fact the compressibility of the gel. The authors should explain how they estimate this parameter instead of just citing another work (reference 48).
- d. It looks like the available number of motors and filaments in the bulk of the droplet are considered to be fixed. Can the authors justify this by estimating what fraction of microtubules/motors are in the cortex relative to the bulk?

6. ATP dependence The authors should check the effect of varying motor activity through ATP concentration in their experiment. This can then allow a verification of the theoretical prediction in Ref. 40 that droplet velocity increases linearly with motor activity.

7. Difference between fig.2 and 3 and corresponding supp movies 2 and 3: same conditions, but 3

shows a clear accumulation of material in the sink on the droplet boundary whereas 2 seems to have homogeneous surface density. Can the authors explain this difference? Related to that, why don't the authors visualize the kinesin in Movie 3? This should prove that kinesin is localizing within the cortex leading to a gradient in active stress, which is the basic hypothesis of the authors' model.

8. Varying surface tension of droplets. Can the authors comment on what determines the equilibrium surface tension of their droplets? One of the possible effects of motor activity in addition to droplet motion is droplet deformation (shape change) by motor-induced mechanical forces. Do the droplets in this experiment remain perfectly spherical, or are there visible deformations? I expect that these deformations can be increased by reducing surface tension.

9. Microtubule turnover. Are MTs taxol stabilized, or do they grow and shrink? If so, can the authors include MT turnover in the model?

10. Citations The introduction/discussion in the paper should benefit from these references:

a. One of the early demonstrations of aster formation in microtubule-motor mixtures:

Surrey et al. *Science*. Vol 292, Issue 5519 pp. 1167-1171

DOI: [10.1126/science.105975](https://doi.org/10.1126/science.105975)

b. Active cytoskeletal liquid crystal droplets also spontaneously formed from entropic interactions:

Weirich et al. *PNAS* 116 (23) 11125-11130 (2019), and related model which captures aster formation within droplets: Schwarzendahl et al. *Physical Rev. Research* 2021 DOI:

[10.1103/PhysRevResearch.3.043061](https://doi.org/10.1103/PhysRevResearch.3.043061)

c. Active microtubule networks and depletion forces:

Nasirimarekani et al. *Langmuir* 2021, 37, 26, 7919-792

d. a suitable review article on self-propelled droplets e.g. Maas et al.

<https://doi.org/10.1146/annurev-conmatphys-031115-011517>

e. active contractility based motility model, similar to what authors use: Nickaen et al. *PLoS Comput Biol* 13(11): e1005862. <https://doi.org/10.1371/journal.pcbi.1005862>

Minor points:

1. State the definition of ρ and the meaning of the color coding in Fig. 2E

2. Improve or clarify the notation used in Eqs. 1-3. For example, it is extremely unclear why the authors need to define $\hat{\rho}_r$ and $\hat{\mu}_r$, and what they physically mean. The C_K and C_M seem sufficient.

Reviewer #3 (Remarks to the Author):

The authors create microdroplets from one of the most famous polymer system undergoing phase separation in aqueous solution (Dextran and PEG), that are known to enrich proteins. Here, microtubules are incorporated, which form a cortex-like structure at the droplet interface due to their long persistence length and depletion interactions. Addition of molecularly engineered kinesins and ATP crosslinks the microtubule network and generates forces that lead to aster formation. At certain microtubule:kinesin ratios, the symmetry is broken, which causes vortex flows. The authors investigate the phase space of protein concentrations that generate flows experimentally as well as by

a 2D mathematical model. They further claim that vortex flows and friction with the surrounding of droplets generate directed motion of droplets, which could be used as meso-scale actuator, however this is not well supported by the data.

The presented system shows features resembling biological processes, such as self-assembly of higher order structures, symmetry breaking in active matter and emerging flows, that are interesting to study with such minimal components due to the ease of reconstruction and could inspire further research in the field. However, cellular microtubule networks look remarkably different and the cellular cortex is composed of actin-myosin instead. Therefore, the claims in the introduction and discussion including modelling intracellular condensates, whole moving cells and assembly of protein motors seem to be overstated. Overall, the manuscript is hard to follow as it is lacking clear language, structure, experimental details and analysis of results, which require substantial rewriting of large parts. Note, that I am not qualified to judge the mathematical model, but I have a number of major and minor concerns that would need to be addressed before publication:

Major points:

- 1) The presented data of one droplet moving through a field of view that is lacking any reference, does not convincingly supporting the claim that "droplet showed translational motion in the sink direction". In the movie also large-scale flows of the medium and fusion events are observable that question the result. Statistical analysis and control experiments without motors/ATP as well as experimental details would be needed to support the claim.
- 2) Please give enough experimental details to understand the set-up. Friction and confinement of droplets are mentioned but it remains elusive how the confinement looks like and which interfaces cause friction. Also have MTs been "added" for data in Fig. 4b? Further details on protein production and image analysis are lacking (are these confocal z-projections – or images from a particular plane?)
- 3) Statistical analysis on any of the reported numbers is missing. How many droplets have been observed, in how many repeats of the experiment, what is the spread of the flow speed ("around 50 nm/s")?
- 4) The orientation of the sink/source should be random in each individual droplet assuming spontaneous symmetry breaking. However, from the pictures/movies neighboring droplets seem to align. Please comment on this.
- 5) The text seems to miss the results of the PIV analysis (Fig 2b).
- 6) How can the flow regime of low CM/low Ck in the experimental phase diagram (Fig. 4a) be explained, that is not captured by the model?
- 7) Introduction and results are not well structured. Some detailed results and methods are anticipated ("as shown in Figure 1a") in the introduction, while not discussed in the results part anymore. Background information and literature on the system, however, would fit better into the introduction than in results ("this type of PS ...has been well studied..."). The introduction ends with discussion of their observations and comparison to literature ("this mesoscale structure is in clear contrast to the well-known MT-kinesin system...") that should be tackled in discussion section and rather end with a brief summary of the most important findings here. Also referring in the conclusion to further unrelated studies (43) is not helpful and should be moved to the introduction section.
- 8) The cited literature often seems irrelevant and not supporting the main message of the sentence e.g. 1, 11, 12, 14 (Brangwynne et al. showed exactly the opposite: no flows needed!) 28-30. Please go through carefully and add original papers or relevant reviews where appropriate (e.g. for Darcy's law).
- 9) The paragraph on membraneless compartments and cytoskeletal proteins in the introduction needs to be carefully rewritten as a whole. E.g. active droplet systems have been studied before (e.g. Donau, C. et al. Active coacervate droplets as a model for membraneless organelles and protocells. Nat. Commun. 11, (2020)., Zwicker, D., Hyman, A. A. & Jülicher, F. Suppression of Ostwald ripening in active emulsions. Phys. Rev. E - Stat. Nonlinear, Soft Matter Phys. 92, 1–13 (2015)) and it is not clear which aspects of membraneless compartments are modelled here and what consequences this would have for the field (maybe the study of Böddeker et al (2022) Nat Physics could be a link). Also why do the droplets here do not undergo ripening and coarsening as phase separated droplets do but

rather “keep their size” even in absence of active components?

Minor points:

10) Wording:

a. While “water/water phase separation” is technically a correct term and might be used to explain the system in the introduction, the PEG/dextran system has been employed for decades and commonly termed “aqueous two-phase system”, which should be mentioned to guide the reader to the relevant literature (e.g. cite in introduction: P.-Å. Albertsson, Wiley, New York (1986), H. Walter, D.E. Brooks and D. Fisher Academic Press, Orlando (1985) Aqueous Two-Phase Systems: Methods and Protocols, R. Hatti-Kaul, Humana Press, Totowa, N.J. (2000), W.A. Aumiller C. D. Keating (2017) Advances in Colloid and Interface Science) Also “(cell-sized aqueous) micro droplets”, as introduced by the authors previously (ChemBioChem 2018) or simply “Dextran-PEG droplets” could be an option for the title and throughout the manuscript that makes it easier for the reader and sound less overselling.

b. While “LLPS” is a physical mechanism that has been proposed to aid the formation of membraneless compartments in cells, care has to be taken since not all condensates are formed via this mechanism.

c. The methodology should be explained in detail and it is fair to stress the simplicity of the approach. However, the repeated usage of the term “simple mechanical mixing” raises the question, how their approach is different from all previous PEG/dextran systems with partitioned proteins?

d. “Compartmentation” should be compartments or compartmentalization

e. When giving relative sizes, such as large, high or low, it would be beneficial to know what is the reference. E.g. actin filaments are generally regarded to have short persistence length in contrast to other cytoskeletal polymers. For protein concentrations in the study give actual numbers or their ratios instead of high/low.

f. What is vortical flow “inside and outside of droplets”?

11) References to the movies are lacking.

12) While motility is a great feature, not all parts of “living systems” need active transport. E.g. it is unclear to me why “transport [is] necessary for their [condensate] activity”, while a hallmark of membraneless condensates is their passive formation despite being in out-of-equilibrium environment. Many of their functions such as storage or reaction sites are independent of active transport!

13) In the abstract the “potential for the assembly of molecular machines, including protein motors” is mentioned. Please specify what is meant with this and how this relates to the presented study.

Point-by-point response to the referees' comments

Referee's comments:

Referee 1

In the manuscript entitled "Self-Emergent Vortex Flow of Microtubule/Kinesin in Cell-Sized Droplets under Water/Water Phase Separation," the authors describe a study in which they merge microtubules and molecular motors with a phase separating two-phase PEG/Dextran system.

The authors describe interesting experimental observations in which the cytoskeletal proteins strongly partition to the Dextran-rich phase. When confined within such a droplet the cytoskeletal components generate autonomous flows. In principle, these are interesting observations that could have an impact on the field of active matter. With that being said I do believe that some of the observations need to be described/quantified in more detail before the manuscript is acceptable for publication.

Response: We would like to thank the referee for the careful review and encouraging comments on our work. Our point-by-point replies and the modifications on our revised manuscript are listed below.

1. The authors claim that microtubules form an aster-like structure when confined within Dextran droplets. Asters have an outwardly splayed structure with motor proteins concentrated at their center. The authors need to demonstrate the presence of such structures, as has been done previously in uniform one-component systems. This could be accomplished by using high numerical aperture water immersion objectives as has been done for numerous other cytoskeletal structures.

Response: To address this, we performed additional experiments investigating the aster formation in individual w/wPS droplets. As shown in Fig. S1, we observed the typical features of microtubule asters. Microtubules spread outward and kinesin molecules concentrated in their centre. These asters eventually increased in size before the vortical flow started. This initial behaviour associated with aster formation and growth is consistent with the observation on the formation of aster-like structure in a homogeneous one-component system reported in Ref. 39 (Torisawa, *et al.*).

The explanation is provided on Page 8, Line 10.

We confirmed the formation of aster structure within a droplet, and the subsequent aggregate formation was confirmed via the bulk experiment with the presence of DEX-PEG (Fig. S1c, d).

Fig. S1c, d, the representation of the aster formation in a droplet has been added to the supplementary information.

Furthermore, we have added the schematic illustration of the aster structure in Fig. S1b.

2. The authors claim that vortex flows can translate droplets. This is an intriguing observation, albeit one that has been observed in several other cytoskeletal systems at significantly larger speeds. However, one needs to demonstrate a stronger connection between cytoskeletal structure, the flows they cause, and the resulting droplet motion. In particular, it would be helpful to simultaneously study the activity-powered motion of many droplets in one field of view to demonstrate that their directions are uncorrelated to each other. The motility of droplets is sufficiently slow that one needs to convincingly eliminate other possible causes of drift which could be external flows or even spatial changes in droplet confinement when the droplet diameter is smaller than the chamber thickness.

Response: We agree with the referee's opinion that it is necessary to eliminate possible drift causes. To this end, we conducted additional experiments to analyse the translation of droplets in one field of view and discovered that there is a correlation between the directions of droplet translation and the positioning of cap-like aggregate structure of the droplet as shown in Fig.3c. Since the cap-like structure always appears in the sink direction of the vortex flow in the droplet, the direction of translation coincides with the sink direction of a pair of vortices. Conversely, the translational directions of droplets were uncorrelated to each other in a given field of view. Such non-coherent directionality of the translation dismisses the possibility that external forces, such as gravity, are the cause of drift motion. In the revised version of the manuscript, we have provided the explanation at L.11 on P. 9.

Furthermore, the droplets were shown to exhibit a translation in the sink direction characterized by the cap-like structure of the aggregate (Fig. 3a, Supplementary Movie S3). The typical speed of the droplet motion was 42.1 ± 7.4 nm/s ($N = 11$). This is comparable to the velocity of the vortex flow; however, an order of magnitude slower than the velocity of the kinesin on the filament³⁹ (Fig. 3b). The direction of the droplet translation in the field of view is different for each droplet and does not give apparent correlation. Conversely, the translation direction of an individual droplet correlates with positioning of the cap-like structure in the droplet. As the position of the cap-like structure indicates the sink direction of the vortex flow inside the droplet (Fig. 3c), the droplet translation should be driven by the vortex flow. These observations rule out the possibility of drift motion caused by external forces such as gravity.

Referee2

The authors Sakuta et al. show that droplets spontaneously formed by depletant-induced phase separation can be used to enclose cytoskeletal gels (microtubule-kinesin mixtures). Interestingly, the motor activity leads to active behavior of the droplet leading to vortical flows within the droplet and their subsequent translation. Based on the observed flows of microtubule in the droplet, the authors provide a minimal mechanical model based on active gel theory that captures how droplet activity depends on a competition of motor and microtubule concentration.

There is considerable interest in reconstituted systems that exhibit life-like properties such as the ability to spontaneously move, in materials science and synthetic biology. As such, this work is a welcome addition to the considerable literature on strategies to create self-propelling droplets. The ability to enclose cytoskeletal gels in a simple type of water/water phase-separated droplet is certainly exciting.

However, there are important questions that the present study does not explore or answer (detailed below), which I think are imperative for a complete scientific study. These need to be addressed before the manuscript can be considered for publication.

Response: We thank the referee for evaluating the life-like characteristics of our system. We revised the manuscript by addressing the questions raised.

1. Aster formation is not directly demonstrated/explained. The authors say they expect asters based on their previous work, “As confirmed in a previous study³², this combination of MTs and the kinesin rapidly forms aster structures”. However, it is not explained what exactly these asters correspond to in the present experiment. Are these the bright clusters of MTs and kinesin seen in the bulk of the droplet? Or do they mean the aggregation of MTs in the cortex, which is really what they use in their computational model?

a. How do we know these asters are contractile based on the evidence presented in this paper?

b. Can the authors visualize the radial alignment of the microtubules in these asters?

Response: We observed the aster formation and aggregation *in situ* within a droplet. As shown in Fig. S1, we confirmed the formation of the asters in which MTs extended radially from the kinesin accumulating centre. The contractility was also confirmed by the observation that two asters interacting with elongated MTs began to aggregate into one large aster, as shown in Fig.S1c. The asters eventually increased their size before the vortical flow started. The initial behaviour underlying the formation and growth of such aggregates was previously described and numerically modelled by our group (ref. 39, Torisawa, *et al.*). The aster structure and the subsequent aggregate formation is attributed to radially aligned MTs with the minus end distally. A schematic representation has been added to Fig.S3b to aid the readers' understanding.

To incorporate the above explanations related with the *in-situ* measurement of aster formation, in the revised version of the manuscript we have added the following sentence at the L. 10 on P. 8 as:

We confirmed the formation of aster structure within a droplet, and the subsequent aggregate formation was confirmed via the bulk experiment with the presence of DEX-PEG (Fig. S1c, d).

We have also added a sentence explaining the contractile nature of the aster structure based on the reference (Ref. 39) at the L. 12 on P. 6 as:

Asters bridged by kinesin (i.e., aster-kinesin complexes) create a contractile network as schematically shown in Fig. S1b, a scheme that has been confirmed numerically.

c. What sets this MT-kinesin system apart from the active nematic system which gives extensile as opposed to contractile flows?

Response: The tetrameric motor used in this study is a chimera composed of four motor domains of conventional kinesin and the Eg5 tails (4-headed KIF5B_{head}-Eg5_{tail}). It moves faster and generates a higher force than that of conventional Eg5 (Ref 39 Torisawa, *et al.*). Upon mixing the motor with microtubules, the motor forms an aster structure with minus-end of the microtubule spreading outward from the centre. The motors free in solution can interact and crosslink the microtubules extending from the asters and then pull these asters towards each other leading to their contraction. Thus, this aster polarity enables system contractility. A previous study from our research group (Ref. 39) confirmed the emergence of the contractile network both experimentally and numerically. In addition, the theoretical study shows that the active stress can be transiently contractile when the cross-linker turn over; thus, leading to cluster formation (Role of Turnover in Active Stress Generation in a Filament Network, T. Hiraiwa, *et al.*, *Phys. Rev. Lett.* (2016) DOI: 10.1103/PhysRevLett.116.188101). To aid understanding, we have revised the beginning of the Results and Discussion at the L. 4 on P. 6 as:

Aster formation of MTs and chimeric kinesin construct, 4-headed KIF5B_{head}-Eg5_{tail}

In this study, we used a mixture of MTs and a member of the kinesin family, 4-headed KIF5B_{head}-Eg5_{tail}, which locally produce active contractile stress, as in the actomyosin system^{34,35}. As confirmed in our previous study³⁹, this combination of MTs and the kinesin rapidly forms aster structures within 5 min, when they are mixed under conditions appropriate for motility (Fig. S1a) in the absence of DEX and PEG. An aster is a dynamic structure consisting of a radial array of MT with a node at its centre, where kinesin is concentrated. The plus-end of the MT points inward, in the direction of the node that is accumulated by the kinesin activity. Asters bridged by kinesin (i.e., aster-kinesin complexes) create a contractile network as schematically shown in Fig. S1b, a scheme that has been confirmed numerically. Such a mesoscale structure is in clear contrast to the well-known MT-kinesin system that generates extensile stress⁴⁰.

We have also added the description of the *in-situ* observation of the aster structure in the composition of the same droplet system at L. 9 of the P. 8 as:

This duration of the induction period correlates well with the time required for aster formation. We confirmed the formation of aster structure within a droplet, and the subsequent aggregate formation was confirmed via the bulk experiment with the presence of DEX-PEG (Fig. S1c, d). The observed growth of the size of MT/kinesin aggregates also supports that the created protein complex produces contractile stress.

2. Reason for microtubule aggregation in droplets. The authors write “Depending on their chemical structure, biopolymers spontaneously accumulate in either the droplet’s internal DEX-rich phase or its external PEG-rich phase”. This requires further explanation. What about the “chemical structure” of microtubules determines that it associates with the DEX-rich droplet? (Is it size, charge, or something else?) Can microtubules also associate with PEG under other experimental conditions?

Response: To clarify the physico-chemical meaning of w/wPS, we have added brief descriptions of the essential characteristics of w/wPS.

In L. 12 on P. 3 as:

In the present article, we adopted the term w/wPS to interpret the formation of aqueous microdroplets surrounded by aqueous solution, which is a cause of aqueous two-phase system (ATPS) often used in chemical engineering^{9,11-13}. Aqueous solution of dextran (DEX) and polyethylene glycol (PEG) is a typical example of ATPS and w/wPS¹⁴⁻¹⁸. Both DEX and PEG are hydrophilic polymers that are readily soluble in water. However,

their mixture induces w/wPS due to a reduced contribution of conformation entropy for polymer solutions (i.e., driven by depletion interaction). One of the characteristics of w/wPS is its ultralow interfacial tension whose order ranges from 1 to 100 $\mu\text{N/m}$ ¹⁹. This low interfacial tension allows the droplet coalescence to slow down and produce a long-lived micrometre scale droplet. Variation of partitioning ability of chemicals are another feature of w/wPS. Aqueous chemicals and biopolymers may be distributed homogeneously in both the phases, one of the phases, or at the interface of the w/wPS. These distributions can be controlled by pH, additional ions, and molecular size. Such characteristics are the foundation of the ATPS method based on w/wPS.

In L. 1 on P. 5

Here, short-strand DNA and G-actin were distributed in both PEG-rich/DEX-rich phases, while long-strand DNA and F-actin preferred DEX-rich phases. Furthermore, highly polymerized, long F-actin has been shown to localize at the interface between PEG-rich/DEX-rich phases¹⁷. This sensitivity of the partitioning to the molecular size is attributable to the different manner of macromolecule packing between the PEG-rich and DEX-rich phases caused by the depletion effect owe to the large difference of polymer flexibility²⁵ and unique characteristics of w/wPS droplet.

3. No exploration at all of droplet self-propulsion I am missing an explanation in the manuscript of how the droplet translational velocity depends on motor activity and why it moves in the way it does. The authors write, quite cryptically, "Typically, the direction of translational motion of an active droplet in a fluid is opposite that in the present system" and a "a comprehensive analysis of the translational motion of the droplet is beyond the scope of this study". While this maybe so, it seems unfair to the reader to leave them hanging without any explanation whatsoever. The authors should at the very least :

a. experimentally quantify how droplet propulsion velocity depends on motor activity (e.g. by varying ATP) or concentration. Discuss what are the important control parameters for droplet velocity.

Response: We agree with your suggestion regarding the experiments with varying ATP concentrations and their importance to relate vortex flow with translation to microscopic motor activity. However, we must consider that high ATP concentrations (1-10 mM) are necessary to maintain motor activity for long periods of time. At low ATP concentrations (<0.1 mM), motor velocity decreases and the propelled motion tends to terminate sooner. It is known that kinesin motor activity depends on ATP concentrations up to 0.1 mM but saturates at higher concentrations (Ref 45, Inhibition of kinesin motility by ADP and phosphate supports a hand-over-hand mechanism, W. R. Schief, *et al.*, *Proc Natl Acad Sci USA* (2004), DOI: 10.1073/pnas.0304369101.) In this study, we use an ATP concentration of 10 mM, which is sufficient to keep the kinesin motor activity stable during the observation period. Using an ATP concentration of 0.1 mM led to ATP depletion in the system before and during the induction period when the aster structure formed, and no vortex motion was observed. Using an ATP concentration of 1 mM, the concentration at which the velocity is saturated, led to no change in vortex behaviour. A feeder system of ATP is needed to maintain the ATP concentration low (Assays for actin sliding movement over myosin-coated surfaces S. J Kron, *et al.*, *Methods in Enzymology* (1991), DOI: 10.1016/0076-6879(91)96035-p). However, employing such a feeder system would require additional proteins and chemicals, which could affect the properties of the DEX/PEG system. It should also be noted that changing the ion concentration in mM units will change the equilibrium distribution ratio (Ref. 13 P.-Å. Albertsson). The main objective of this study is to determine the potential for self-assembly of MT-kinesin molecular motors in w/wPS droplets. In the future subject, it may be interesting to examine the effect of motor activity by changing the concentration of ATP.

In addition, based on the above comment, we realised that we should mention that the ATP concentration was high enough to saturate kinesin activity. We have inserted sentences at L. 5 on P. 8 as:

Here, the ATP concentration used was 10 mM, a concentration 100 times higher than that observed when kinesin activity is saturated⁴⁵.

b. extend the simple model to provide a mathematical relationship between droplet velocity and the important control parameters. This has already been done in ref. 40 by mapping the active cortical flows to Marangoni flows based on an effective surface tension gradient. This predicts a linear relationship between velocity and motor activity.

Response: The model presented in reference 40 (49 in revised manuscript; Singh *et al.*, *Phys. Rev. Res.* (2020)) was developed based on the modelling method with a more general situation in which chemical droplet motion could also be described. For this reason, detailed parameters such as kinesin and MT concentrations cannot be incorporated directly into the model, but are rounded to the activity parameter. Due to the rounding process, the velocity of droplets is proportional to the activity parameter, and hence motor activity with a reasonable approximation.

In contrast, we have extended the "model based on motor and polymer concentration" presented in reference 41 (53 in revised ver.; Hawkins *et al.*, *Biophys J.* (2011)) to reproduce our experiments. In this model, the linear relationship of the activity parameters is implicitly given in Equation 20, and parameters given in the equation can directly reflect the motor activity as well as its concentration, so as to make the correspondence to the experimental conditions. This is an advantage of our modelling and a major difference from the model proposed in reference 40. We believe that our model is rather general and has great generalizability, in spite of its simplicity.

We should also note that the vortical flow generation and the translational motion of the droplet requires additional consideration. The direction of the translational motion is opposite to the conventional swimming droplet, probably due to the strongly confined nature of w/wPS droplet by the substrate. Further hydrodynamic consideration by three-dimensional model will be expected as a future study. We have added a sentence at L. 14 on P. 13 as:

We also identified several research questions that should be investigated in future studies. First, our experimental observation that the vortex core line gradually tilts to become parallel to the substrate with an increase in the kinesin concentration should be validated. Next, the relation between the directions of vortex flow and motion should be investigated in detail. Explaining these observations by extending the model for a 3-dimensional flow analysis^{53,56} by adopting observed droplet shape (Supplementary Fig. S2) would add significant value to the existing scientific information.

c. discuss qualitatively the reason for the droplet movement in a certain direction

Response: We have revised the description of the droplet movement accordingly. The direction of the translational motion as well as the sink of the vortical flow were randomly chosen. This fact eliminates the possibility that the translational motion of the droplet is due to an external force such as gravity. Furthermore, the directions of the translational motion and the sink were aligned, which proves the motion is due to the vortical flow in the droplets. Thus, our mathematical model suggests the appearance of vortical flow is based on the spontaneous symmetry breaking, and the direction is randomly chosen reflecting small disturbance and/or fluctuation in the initial conditions.

To show these results, we have added Fig. 3c to the revised manuscript, and have inserted sentences at L. 15 on

P. 9 as:

The direction of the droplet translation in the field of view is different for each droplet and does not give apparent correlation. Conversely, the translation direction of an individual droplet correlates with positioning of the cap-like structure in the droplet. As the position of the cap-like structure indicates the sink direction of the vortex flow inside the droplet (Fig. 3c), the droplet translation should be driven by the vortex flow. These observations rule out the possibility of drift motion caused by external forces such as gravity.

The initially random vortex direction is gradually aligned between droplets at the later stage as shown in Fig. 1c in the modified manuscript (1a in the previous manuscript) as well as in the supporting movie S1. Such tendencies are observed quite often and should underlie interaction between droplets via external continuous PEG-rich phase. These hydrodynamics as well as chemical interactions are of interests especially when we consider the multiple droplet systems often discussed in the active matter physics community. In the revised manuscript, we have addressed this issue as a possible future study in L. 9 on P. 10 as:

We note that the vortex directions within neighbouring droplets synchronize with each other (Fig. 3c). Such coupling of flow inside of droplets is probably possible by the concentration of ATP in the continuous phase and would be characteristic of w/wPS droplets where all phases are aqueous. Furthermore, the coupling may trigger coherent droplet motion that would be interesting to investigate especially in terms of collective behaviour of droplets.

4. Discrepancy between model and experimental phase diagrams. There is an obvious difference between experimental observation and the model prediction (Figs. 4a and 4c). The theoretical prediction is a linear relation between C_K and C_M , but the experiment deviates from that at low C_M . The authors have not commented at all on this, but it is very important that they at least discuss this and the possible drawbacks in their model.

Response: We used the simple mathematical model to exemplify the essence of vortex appearance in a droplet due to the contractile MT-kinesin structure. The model can predict well the overall behaviour of the system. However, its characteristic simplifications led to discrepancies in experimental data with the low C_M range. This was attributed to the fact that the model overestimated hydrostatic pressure in low C_M region and/or the underestimation of contractile stress at low C_K range. The manuscript has been revised and a new description has been added at L. 8 on P. 13.

In the region with low C_K and C_M , the prediction of our model deviates from the experimental results. This discrepancy could be due to the simplicity of our model, which overestimated the hydrostatic pressure at low C_M and/or underestimated the contractile stress at low C_K . Nevertheless, the essential picture of droplet motion as well as semi-quantitative agreement with the experiment were established.

5. Statement of assumptions in the model. There are several unstated assumptions in the simple model that the authors use, for example:

- a. viscosity within the cortical layer is ignored, and the fluid is treated as overdamped.*
- b. possible polarization and anisotropy of the microtubule distribution and active stresses are ignored.*

Response: We have added these descriptions in the main text at L. 5 on P. 12, as follows:

For simplicity, we neglect the viscosity in the cortex and assume the fluid motion is overdamped. In addition, a

possible polarization and anisotropy of aster structure in the cortex is neglected; thus, the generated stress is assumed to be isotropic.

We have added the following explanation in the method section at L. 12 on P. 17:

For simplicity, we ignored the viscosity within the cortex. Furthermore, the possible polarization and anisotropy of aster structure in the cortex was neglected. Therefore, we assumed that the generated stress is isotropic.

c. The origin of the hydrostatic pressure is actually the resistance of the gel to deformation, so the parameter α is in fact the compressibility of the gel. The authors should explain how they estimate this parameter instead of just citing another work (reference 48).

Response: Indeed, α is the inverse of the gel compressibility, and we estimated its order of magnitude using the value of f-actin given in reference 48 (60 in revised ver.; Wagner *et al.*, *Biophys J.* (1999)). To clarify this point, we have added a sentence in the method section at L. 16 on P. 21, as follows:

For α , which is the inverse of the adiabatic compressibility of aster, we estimated its order from those of F-actin⁶⁰, $6.9 \times 10^{-10} \text{ Pa}^{-1}$, and used $\alpha = 10^8 \text{ Pa}$, which is slightly smaller, and hence softer, for considering the sparse aster structure in the present study.

d. It looks like the available number of motors and filaments in the bulk of the droplet are considered to be fixed. Can the authors justify this by estimating what fraction of microtubules/motors are in the cortex relative to the bulk?

Response: Thank you for pointing out this important issue. The motors and asters in the bulk gradually decrease their number after approximately an hour. At this late stage, the aggregate creates a cap-like structure in the sink direction accumulating most motors and asters. Thus, the available number of proteins in a single droplet is run out in approximately of the order of 100 min. In our mathematical model, we focus our attention on the initial regime of the flow structure in the droplet (corresponding initial few minutes, where the concentration of bulk remains almost constant.). In the initial stage, we assumed the available concentration of motors and asters in the bulk to be fixed. This point should be stated in the main text. We, therefore, have added a sentence in the main text at L. 4 at P. 12:

We focus on the initial regime of the flow structure in the droplet (corresponding initial 10 min).

We have also added the following explanation in the method section at L. 7 on P. 17:

Notably, the motors and asters in a single droplet gradually decreased their number after approximately one h by creating the accumulated cortex at the interface. We focused on the initial regime of the flow structure in the droplet (corresponding initial 10 min). In such an initial regime, we assumed that the available concentration of motors and asters in a droplet is fixed and constant.

6. ATP dependence The authors should check the effect of varying motor activity through ATP concentration in their experiment. This can then allow a verification of the theoretical prediction in Ref. 40 that droplet velocity increases linearly with motor activity.

Response: As mentioned in the comment at 3a and b, varying the ATP concentration affects many other

properties of the experimental system besides change in motor activity ζ . Instead, we varied the concentration of kinesin to modify effective motor activity. These results were successfully reproduced as in the theoretical discussion, even without the theoretical consideration of ref. 40 (49 in revised ver.; Singh *et al.*, *Phys. Rev. Res.* (2020)).

7. Difference between fig.2 and 3 and corresponding supp movies 2 and 3: same conditions, but 3 shows a clear accumulation of material in the sink on the droplet boundary whereas 2 seems to have homogeneous surface density. Can the authors explain this difference? Related to that, why don't the authors visualize the kinesin in Movie 3? This should prove that kinesin is localizing within the cortex leading to a gradient in active stress, which is the basic hypothesis of the authors' model.

Response: We revised the indicated sections. Fig. 2c was intended to show that the accumulation of motors and filaments in the sink direction with time, creating a cap-like structure at the boundary. However, the kymographic representation is often hard to understand without the presence of snapshots. To clarify these issues we have added snapshots in Fig. 2c of the modified manuscript. Furthermore, as the referee suggests, kinesin is distributed almost homogeneously within the cortex as shown in Supplementary Movie S3 (newly added the kinesin fluorescent image) and indeed this formation of cap-like structure at the cortex with MT and kinesin provide a strong support to our model with gradient in active stress. We have mentioned this at L. 3 on P. 13 as:

The calculation predicts that the steady profile of MT density at the surface breaks symmetry with its peak at the sink flow direction. This is consistent with the experimental observation where a cap-like structure was generated.

8. Varying surface tension of droplets. Can the authors comment on what determines the equilibrium surface tension of their droplets? One of the possible effects of motor activity in addition to droplet motion is droplet deformation (shape change) by motor-induced mechanical forces. Do the droplets in this experiment remain perfectly spherical, or are there visible deformations? I expect that these deformations can be increased by reducing surface tension.

Response: The coupling between motion and deformation is an interesting topic. The w/wPS droplet is characterised by extremely low interfacial tension, of the order of 10^{-6} N/m, which is 10^{-7} of the typical value of interfacial tension between ordinary fluids. With the typical length scale of the present system to be 10^{-6} m, crudely speaking the typical force by the interface is 10^{-12} N, which is the same order of the magnitude as the typical force generated by a single kinesin motor. Indeed, the recently published study (Ref. 52, Adkins *et al.*, *Science* (2022)) shows intensive deformation of the interface. However, we could not identify noticeable deformation due to the flow within a droplet as shown Fig. 1, 2, 3, and Fig. S2, where 3D imaging is depicted. We expect this to be explained by the contractile nature of our combination of MT and kinesin variants. Revealing the mechanism underlying such differences represents a fascinating topic for future study. We have therefore added a comment at L. 14 on P. 10 as:

The coupling between motion and deformation is also an interesting topic especially in terms of active matter physics⁵¹. W/wPS droplets have extremely low interfacial tension, of the order of 10^{-6} N/m, which is 10^{-4} of the typical interfacial tension value observed between ordinary fluids. Although a recently published study⁵² showed intensive deformation of the interface, we could not identify noticeable deformation due to the flow within a droplet as shown in Fig. 1, 2, 3, and Fig. S2, where 3D imaging is depicted. We expect this to be explained by the contractile nature of our combination of MT and kinesin variants. Revealing the mechanism underlying such differences represents a fascinating topic for future study.

9. Microtubule turnover. Are MTs taxol stabilized, or do they grow and shrink? If so, can the authors include

MT turnover in the model?

Response: We used Taxol (paclitaxel) to stabilize MTs. To make clear this point, in the revised manuscript we have mentioned this in the method section.

paclitaxel (Taxol, Sigma-Aldrich, St. Louis, MO, USA).

We may extend our experimental system to include MT turnover by eliminating Taxol from our system. This extension is beyond the scope of the present paper and would be left for a future study.

10. Citations The introduction/discussion in the paper should benefit from these references:

a. One of the early demonstrations of aster formation in microtubule-motor mixtures: Surrey et al. Science. Vol 292, Issue 5519 pp. 1167-1171 DOI: 10.1126/science.105975

b. Active cytoskeletal liquid crystal droplets also spontaneously formed from entropic interactions: Weirich et al. PNAS 116 (23) 11125-11130 (2019), and related model which captures aster formation within droplets: Schwarzendahl et al. Physical Rev. Research 2021 DOI: 10.1103/PhysRevResearch.3.043061

c. Active microtubule networks and depletion forces: Nasirimarekani et al. Langmuir 2021, 37, 26, 7919–792

d. a suitable review article on self-propelled droplets e.g. Maas et al. <https://doi.org/10.1146/annurev-conmatphys-031115-011517>

e. active contractility based motility model, similar to what authors use: Nickaen et al. PLoS Comput Biol 13(11): e1005862. <https://doi.org/10.1371/journal.pcbi.1005862>

Response: Thank you to the useful suggestions. We have added the references and clarifications you suggested.

Minor points:

1. State the definition of ρ and the meaning of the color coding in Fig. 2e

Response: We have inserted the following definition in the caption of Fig. 2:

The colour indicates the MT surface density ρ . The accumulation of ρ is clearly observed

2. Improve or clarify the notation used in Eqs. 1-3. For example, it is extremely unclear why the authors need to define $\hat{\rho}_r$ and $\hat{\mu}_r$, and what they physically mean. The C_K and C_M seem sufficient.

Response: We realized that the introduction of $\hat{\rho}_r$ is redundant. At the same time, we decide to clarify the description of the model. Thus, we have inserted the following sentences at L. 7 on P. 18 as:

Eq. (1) and (2) describe the adsorption equilibrium of MT (kinesin) between bulk and the interface. The diffusion of MT is neglected while those of kinesin is introduced in Eq. (2); thus, reflecting the large difference in molecular size associated with the inclusion of the diffusion term. Eq. (3) describes the active contraction by the first term, the hydrostatic pressure of MT by the second term, where the third term induces the aggregating behavior of MT. Further, ρ_0 and μ_0 represent the reference density of MTs and kinesins (see **Parameter estimation**).

The interpretation on the parameter estimation section has been improved at L. 16 on P. 21 as:

For α , which is the inverse of the adiabatic compressibility of aster, we estimated its order from those of F-actin⁶⁰, $6.9 \times 10^{-10} \text{ Pa}^{-1}$, and used $\alpha = 10^8 \text{ Pa}$, which is slightly smaller, and hence softer, for considering the

sparse aster structure in the present study. We also used $D = 10^{-12} \text{ m}^2/\text{s}$ ^{59,61,62}. We obtain $\hat{\kappa} = 1$, $\hat{D} = 0.001$, and $\hat{\alpha} = 1$. At MT concentration in a droplet $C_{M0} = 1.8 \text{ } \mu\text{M}$, we assumed adsorption equilibrium density $k_+ C_{M0}/k$, coincide with reference density ρ_0 and recognized as the standard condition with active vortical flow. We can rewrite $C_{M0} = \frac{k\xi R_0^2}{\alpha} \frac{k\rho_0}{k_+}$. Note that $\frac{k\xi R_0^2}{\alpha}$ is dimensionless and considered as unity in our parameter estimation.

Referee 3

The authors create microdroplets from one of the most famous polymer system undergoing phase separation in aqueous solution (Dextran and PEG), that are known to enrich proteins. Here, microtubules are incorporated, which form a cortex-like structure at the droplet interface due to their long persistence length and depletion interactions. Addition of molecularly engineered kinesins and ATP crosslinks the microtubule network and generates forces that lead to aster formation. At certain microtubule: kinesin ratios, the symmetry is broken, which causes vortex flows. The authors investigate the phase space of protein concentrations that generate flows experimentally as well as by a 2D mathematical model. They further claim that vortex flows and friction with the surrounding of droplets generate directed motion of droplets, which could be used as meso-scale actuator; however this is not well supported by the data.

The presented system shows features resembling biological processes, such as self-assembly of higher order structures, symmetry breaking in active matter and emerging flows, that are interesting to study with such minimal components due to the ease of reconstruction and could inspire further research in the field. However, cellular microtubule networks look remarkably different and the cellular cortex is composed of actin-myosin instead. Therefore, the claims in the introduction and discussion including modelling intracellular condensates, whole moving cells and assembly of protein motors seem to be overstated. Overall, the manuscript is hard to follow as it is lacking clear language, structure, experimental details and analysis of results, which require substantial rewriting of large parts. Note, that I am not qualified to judge the mathematical model, but I have a number of major and minor concerns that would need to be addressed before publication:

Response: Thank you for your fruitful comment. We have revised our manuscript and provided additional experimental data that support the formation of asters as well as integrated detailed observations of translational motion. Our manuscript revisions and point-by-points replies are provided below.

Major points:

1) The presented data of one droplet moving through a field of view that is lacking any reference, does not convincingly supporting the claim that “droplet showed translational motion in the sink direction”. In the movie also large-scale flows of the medium and fusion events are observable that question the result. Statistical analysis and control experiments without motors/ATP as well as experimental details would be needed to support the claim.

Response: To clarify the translational motion of the droplets generated by the vortical flow inside of the droplet, a wider area was observed, and the motion of multiple droplets was measured. The results show that the translational direction of the droplets were random, but the translational direction of individual droplets was highly correlated with the sink direction of the vortex. This eliminates the possibility that the droplet motion is due to the external forces such as gravity. A figure showing the direction of motion and sink direction has been added at Fig. 3c, and the additional explanations have been added at L. 11 on P. 9 as:

Furthermore, the droplets were shown to exhibit a translation in the sink direction characterized by the cap-like structure of the aggregate (Fig. 3a, Supplementary Movie S3). The typical speed of the droplet motion was 42.1 ± 7.4 nm/s ($N = 11$). This is comparable to the velocity of the vortex flow; however, an order of magnitude slower than the velocity of the kinesin on the filament³⁹ (Fig. 3b). The direction of the droplet translation in the field of view is different for each droplet and does not give apparent correlation. Conversely, the translation direction of an individual droplet correlates with positioning of the cap-like structure in the droplet. As the position of the cap-like structure indicates the sink direction of the vortex flow inside the droplet (Fig. 3c), the droplet translation should be driven by the vortex flow. These observations rule out the possibility of drift motion caused by external forces such as gravity.

2) Please give enough experimental details to understand the set-up. Friction and confinement of droplets are

mentioned but it remains elusive how the confinement looks like and which interfaces cause friction. Also have MTs been “added” for data in Fig. 4b? Further details on protein production and image analysis are lacking (are these confocal z-projections – or images from a particular plane?)

Response: As mentioned in the water/water microdroplet subsection within the Methods section, the chemical composition was initially prepared and then, mechanical mixing with a vortex shaker produced the microdroplets to be used in the experiment. We did not add the materials at a later time point and the change in the concentration, such as those shown in Fig. 4b, represent differences during the preparation step.

For further clarification, we have added the following explanation at L. 4 on P. 8 as:

When ATP was included in a preparation solution (see Supplementary Table S3 for the detailed concentration), vortical flow was generated inside a droplet. Here, the ATP concentration used was 10 mM, a concentration 100 times higher than that observed when kinesin activity is saturated⁴⁵. We mixed the preparation solution using vortex shaker and the vortex emerged after an induction period ranging from a few to tens of min (Fig. 1c, Supplementary Movie S1).

We inserted the description of the droplet shape accordingly. The droplet has a semi-spherical shape that makes contact with the bottom slide glass. We have added the supplementary Fig. S2 and mentioned it in the main text at L. 7 P. 7 as:

The obtained droplet took a semi-spherical shape having contact with a bottom slide glass as shown in Supplementary Fig. S2.

3) Statistical analysis on any of the reported numbers is missing. How many droplets have been observed, in how many repeats of the experiment, what is the spread of the flow speed (“around 50 nm/s”)?

Response: In the original manuscript, the flow speed was given as the experimental values obtained from the slope of the kymograph as shown in Fig. 2c. To gain more reliable data, we newly analysed the data obtained from PIV from 3 independent droplets. In the revised manuscript, we have replaced the value with the averaged (1.3×10^2 nm/s). We have also added supporting Fig. S3 to show the obtained flow velocity data inside droplets.

4) The orientation of the sink/source should be random in each individual droplet assuming spontaneous symmetry breaking. However, from the pictures/movies neighboring droplets seem to align. Please comment on this.

Response: As mentioned in the comment 1), the orientation of the vortex as well as the translational motion of the droplet is random in one fields of view. However, at a later stage as seen in the Fig. 1c, we observed the tendency of vortex direction to synchronize between droplets. This is an interesting coupling through hydrodynamic effect, but we could not accumulate enough data to claim such synchronization. In the revised manuscript, we have provided a simple description of this tendency at L. 9 on P. 10 as:

We note that the vortex directions within neighbouring droplets synchronize with each other (Fig. 3c). Such coupling of flow inside of droplets is probably possible by the concentration of ATP in the continuous phase and would be characteristic of w/wPS droplets where all phases are aqueous. Furthermore, the coupling may trigger coherent droplet motion that would be interesting to investigate especially in terms of collective behaviour of droplets.

5) The text seems to miss the results of the PIV analysis (Fig 2b).

Response: Thanks for your careful check and we realized that we missed a sentence here, the following modifications have been made:

Particle Image Velocimetry (PIV) analysis⁴⁶ shows that the vortical flow near the interface from source to sink direction indicated in Fig. 2b, as well as strong backflow (sink to source) observed at the central part of the droplet (Fig. 2d).

6) How can the flow regime of low C_M /low C_K in the experimental phase diagram (Fig. 4a) be explained, that is not captured by the model?

Response: As suggested, the discrepancy between our model and the experimental results in the region of low C_M /low C_K was not mentioned. Our mathematical model is a simplified model and while it can explain the essential feature of the appearance of droplet vortices due to the contractile MT-kinesin structure, it is not a complete description. In accordance with the referee's comment referring to this discrepancy between the data and the model, we have added the following section:

In the region with low C_K and C_M , the prediction of our model deviates from the experimental results. This discrepancy could be due to the simplicity of our model, which overestimated the hydrostatic pressure at low C_M and/or underestimated the contractile stress at low C_K .

7) Introduction and results are not well structured. Some detailed results and methods are anticipated (“as shown in Figure 1a”) in the introduction, while not discussed in the results part anymore. Background information and literature on the system, however, would fit better into the introduction than in results (“this type of PS ...has been well studied...”). The introduction ends with discussion of their observations and comparison to literature (“this mesoscale structure is in clear contrast to the well-known MT-kinesin system...”) that should be tackled in discussion section and rather end with a brief summary of the most important findings here. Also referring in the conclusion to further unrelated studies (43) is not helpful and should be moved to the introduction section.

Response: We revised the introduction accordingly. We believe this revision drastically improved the introduction section.

8) The cited literature often seems irrelevant and not supporting the main message of the sentence e.g. 1, 11, 12, 14 (Brangwynne et al. showed exactly the opposite: no flows needed!) 28-30. Please go through carefully and add original papers or relevant reviews where appropriate (e.g. for Darcy’s law).

Response: The citation of studies number 1, 11, 12, and 14 in the previous manuscript was meant to support the idea that external flow is needed to produce the LLPS droplet movement. As the main message of ref. 14 is that the external flow does not contribute to the accumulation of P-granule, we realized that citing these papers may cause unnecessary confusion. The revised manuscript has been modified as follows:

On the contrary, most studies on droplets made through w/wPS have considered them as passive objects incapable of self-propelled motility, while membranous systems have incorporated motility successfully by adopting cytoskeletal and motor proteins²⁶⁻³³.

9) The paragraph on membraneless compartments and cytoskeletal proteins in the introduction needs to be

carefully rewritten as a whole. E.g. active droplet systems have been studied before (e.g. Donau, C. et al. Active coacervate droplets as a model for membraneless organelles and protocells. Nat. Commun. 11, (2020)., Zwicker, D., Hyman, A. A. & Jülicher, F. Suppression of Ostwald ripening in active emulsions. Phys. Rev. E - Stat. Nonlinear, Soft Matter Phys. 92, 1–13 (2015)) and it is not clear which aspects of membraneless compartments are modelled here and what consequences this would have for the field (maybe the study of Bøddeker et al (2022) Nat Physics could be a link). Also why do the droplets here do not undergo ripening and coarsening as phase separated droplets do but rather “keep their size” even in absence of active components?

Response: W/wPS droplet is known for its extremely low interfacial tension ($\sim \mu\text{N/m}$) when the composition is close to critical condition. Under such low interfacial tension, the coarsening process is slowed down to such an extent that even droplets with a 10–100 μm diameter can be observed. Thus, our intended meaning with “membraneless compartment” is different from the one mentioned by the referee. For clarity, we have removed the expression “membraneless compartment” from the present version of manuscript and revised the introduction section.

Minor points:

10) Wording:

a. While “water/water phase separation” is technically a correct term and might be used to explain the system in the introduction, the PEG/dextran system has been employed for decades and commonly termed “aqueous two-phase system”, which should be mentioned to guide the reader to the relevant literature (e.g. cite in introduction: P.-Å. Albertsson, Wiley, New York (1986), H. Walter, D.E. Brooks and D. Fisher Academic Press, Orlando (1985) Aqueous Two-Phase Systems: Methods and Protocols, R. Hatti-Kaul, Humana Press, Totowa, N.J. (2000), W.A. Aumiller C. D. Keating (2017) Advances in Colloid and Interface Science) Also “(cell-sized aqueous) micro droplets”, as introduced by the authors previously (ChemBioChem 2018) or simply “Dextran-PEG droplets” could be an option for the title and throughout the manuscript that makes it easier for the reader and sound less overselling.

Response: As in the indicated reference, the phrase “aqueous two phase system: ATPS” refers to a method to extract solutes from a liquid phase. The phrase “water/water phase separation: w/wPS”, alternatively, refers to a physical phenomenon that leads to ATPS production. We agree that there are numerous relevant studies investigating ATPS; however, the phenomenon we focused on was droplet creation using w/wPS. Therefore, we would like to keep using the term w/wPS. In the revised manuscript, we have tried to provide a guide for readers regarding the potential ATPS confusion at L. 12 on P. 3 as:

In the present article, we adopted the term w/wPS to interpret the formation of aqueous microdroplets surrounded by aqueous solution, which is a cause of aqueous two-phase system (ATPS) often used in chemical engineering^{9,11-13}. Aqueous solution of dextran (DEX) and polyethylene glycol (PEG) is a typical example of ATPS and w/wPS¹⁴⁻¹⁸. Both DEX and PEG are hydrophilic polymers that are readily soluble in water. However, their mixture induces w/wPS due to a reduced contribution of conformation entropy for polymer solutions (i.e., driven by depletion interaction). One of the characteristics of w/wPS is its ultralow interfacial tension whose order ranges from 1 to 100 $\mu\text{N/m}$ ¹⁹. This low interfacial tension allows the droplet coalescence to slow down and produce a long-lived micrometre scale droplet. Variation of partitioning ability of chemicals are another feature of w/wPS. Aqueous chemicals and biopolymers may be distributed homogeneously in both the phases, one of the phases, or at the interface of the w/wPS. These distributions can be controlled by pH, additional ions, and molecular size. Such characteristics are the foundation of the ATPS method based on w/wPS.

b. While “LLPS” is a physical mechanism that has been proposed to aid the formation of membraneless compartments in cells, care has to be taken since not all condensates are formed via this mechanism.

Response: We realized that the explanations for w/wPS and LLPS were not satisfactory and caused confusion. Hence, in the revised manuscript, we have focused the introduction on w/wPS, and briefly mentioned LLPS.

c. The methodology should be explained in detail and it is fair to stress the simplicity of the approach. However, the repeated usage of the term “simple mechanical mixing” raises the question, how their approach is different from all previous PEG/dextran systems with partitioned proteins?

Response: Here we compared the preparation of the DEX/PEG droplet system with the conventional method to prepare droplet using a microfluidics device. Further, mechanical mixing has been rewritten as “mixing using vortex shaker for a few seconds” and we have avoided the redundant use of “simple”.

d. “Compartmentation” should be compartments or compartmentalization

Response: Following the referee’s comment, we have replaced the term “compartmentation” with “compartments”.

e. When giving relative sizes, such as large, high or low, it would be beneficial to know what is the reference. E.g. actin filaments are generally regarded to have short persistence length in contrast to other cytoskeletal polymers. For protein concentrations in the study give actual numbers or their ratios instead of high/low.

Response: We have inserted the actual number of persistent length and concentration of MT at L. 15 on P. 7.

f. What is vortical flow “inside and outside of droplets”?

Response: Indeed, it is inappropriate to use terms such as vortical flow inside and outside of droplets. We have rephrased as follow:

Furthermore, the generated droplets cause the vortical flow inside droplets, which also produces a flow outside the droplets.

11) References to the movies are lacking.

Response: We have added the references to the supplementary movies in the manuscript.

12) While motility is a great feature, not all parts of “living systems” need active transport. E.g. it is unclear to me why “transport [is] necessary for their [condensate] activity”, while a hallmark of membraneless condensates is their passive formation despite being in out-of-equilibrium environment. Many of their functions such as storage or reaction sites are independent of active transport!

Response: We agree that not all condensate activity requires active transport. Here, we meant that a droplet made under w/wPS has been considered to be merely a passive object. This means that an external flow is necessary to introduce long-range transportation, longer than diffusion length $\sim(Dt)^{1/2}$. To clarify our introductory paragraphs, we decided to rewrite the introduction, and have introduced the following section at L. 9 on P. 5 as:

Spontaneous movement, one of the essential features of living systems, is achieved by consuming chemical energy, such as ATP, under isothermal conditions. On the contrary, most studies on droplets made through w/wPS have considered them as passive objects incapable of self-propelled motility, while membranous systems

have incorporated motility successfully by adopting cytoskeletal and motor proteins²⁶⁻³³.

13) In the abstract the “potential for the assembly of molecular machines, including protein motors” is mentioned. Please specify what is meant with this and how this relates to the presented study.

Response: We agree that we should specify the meaning of the potential ability. We have revised the sentence at L. 1 from the bottom on P. 14 as:

Our work reveals that the interface of w/wPS can be used for assembling various molecular machines that can function in an organized manner.

Reviewers' comments:

Reviewer #1 (Remarks to the Author):

The resubmission of the manuscript entitled "Self-Emergent Vortex Flow of Microtubule/Kinesin in Cell-Sized Droplets under Water/Water Phase Separation," addresses some of the concerns raised.

In response to the query about the existence of asters, the authors introduced supplementary figure 1. Here, they demonstrate the existence of aster-like structures in the absence of the polymer. Subsequently, they show behavior in the presence of PEG/Dextran. The authors argue that these images demonstrate the existence of asters in presence of polymer. However, I remain unconvinced. The images show the formation of microtubule-motor aggregate contractile structure, but I would argue that these structures are not asters. Asters by definition have a spherical core enriched in motor proteins with radially splaying outward filamentous microtubules. The images do not show a spherical core and there is no support for the statement that microtubules are pointing radially outward nor that they are polarity sorted. The addition of PEG or Dextran will induce attractive depletion-like interactions between microtubules, which will induce their bundling and is thus likely to significantly affect the behavior of the composite system. This is important as the theoretical model assumes existence of asters.

There are several other statements that need to be justified with quantitative data. Specifically, the authors state that over time microtubules are transported from the droplet interior to the cortex and there they form a cap. This should be quantified by integrating the fluorescence intensity of surface-bound and interior microtubules.

The enthusiasm for the manuscript is somewhat diminished as it seems that the proposed mechanism is transient and the droplets on long time scales become quiescent. The authors claim that examining long-term dynamics is challenging due to the requirement to employ the ATP regeneration system. However, I would note that such a regeneration system has been employed in a recent complementary manuscript (ref 52), and this enabled the study of the dynamics over multiple hours.

Reviewer #2 (Remarks to the Author):

The authors have satisfactorily addressed nearly all reviewer questions in the revised and improved version of the manuscript. I would therefore recommend acceptance with minor revisions.

Two points related to my previous questions that the authors may want to consider:

1. Does the velocity of the droplet motion (stated to be 42.1 ± 7.4 nm/s on page 9) depend on kinesin density? The authors have addressed my question about varying motor activity through ATP concentration, which is apparently not the best knob to turn for the current setup. However, the active stress should depend on motor density as in the model Eq. 3, leading to stronger flows for higher C_K . It is not made clear if (or why not) this is observed experimentally. The authors should also be able to explore this easily with their modeling, but I could not find a reference to the droplet velocity measurement from simulations.

2. The authors now discuss the possible shape changes of active droplets as an important topic for future study. They conjecture that they do not observe such shape deformations in their system because: "We expect this to be explained by the contractile nature of our combination of MT and kinesin variants" Perhaps they can explain this a little more, because it is unclear to me why contractile stresses cannot cause deformations. At least in simulations, contractile stresses have been seen to produce droplet deformation, e.g. Ref. 56 Nickaen et al.

Reviewer #3 (Remarks to the Author):

The manuscript was largely approved and the authors responded satisfactorily to all my comments. One minor comment: the newly inserted sentence on the PIV analysis reads odd and should be revised for clarity. I support the publication of the manuscript and congratulate the authors to their sound work!

Replies to the Referee' comments:

Replies to the comments by Referee 1

In response to the query about the existence of asters, the authors introduced supplementary figure 1. Here, they demonstrate the existence of aster-like structures in the absence of the polymer. Subsequently, they show behavior in the presence of PEG/Dextran. The authors argue that these images demonstrate the existence of asters in presence of polymer. However, I remain unconvinced. The images show the formation of microtubule-motor aggregate contractile structure, but I would argue that these structures are not asters. Asters by definition have a spherical core enriched in motor proteins with radially splaying outward filamentous microtubules. The images do not show a spherical core and there is no support for the statement that microtubules are pointing radially outward nor that they are polarity sorted. The addition of PEG or Dextran will induce attractive depletion-like interactions between microtubules, which will induce their bundling and is thus likely to significantly affect the behavior of the composite system. This is important as the theoretical model assumes existence of asters.

Response:

Following the comment by the referee, we have replaced the term of aster by aster-like in the main text and also in the supplemental Figure, in order to diminish the unsuitable understanding on the term of aster.

In our study, we use the term, aster-like structure, as MT-kinesin complex where many kinesins are accumulated at the central part of radially extended MTs. As previously published in Torisawa, et al., Biophys J. 2016, the structure shows contractile network in the absence of PEG/Dextran, and the polarity of MTs are assumed to be aligned as shown in Supplementary Fig b in the present manuscript. The degree of sorted polarity is not quantitatively estimated, but we confirmed the network is indeed contractile, not extensile. The report in Torisawa Biophys J. has confirmed also that the contractile behavior of the same MT-kinesin complex in the absence of the PEG/Dextran. This fact proves that the depletion effect is not main component for the aggregate formation. In addition, we confirmed the same structure of MT-kinesin where kinesin is accumulated at the core part of the radially extended MTs even in the presence of PEG/Dextran in the supplementary Fig 1. The fluorescence image in the supplementary Fig. 1 reveals the formation of aster-like structure of the MT-kinesin complex.

Based on the above data, it would be reasonable to assume the contractile meso-structure, which we call aster in the previous version. The reproducibility of mathematical model further supports the above assumptions. However, we realized that the observed structure is not perfectly agree with the definition of aster and accordingly decide to modify the expression.

There are several other statements that need to be justified with quantitative data. Specifically, the authors state that over time microtubules are transported from the droplet interior to the cortex and there they form a cap. This should be quantified by integrating the fluorescence intensity of surface-bound and interior microtubules.

Response:

To show the adsorption dynamics of MTs, we have newly added this data on supplementary Fig. S1. We have also added the following sentence in the manuscript on P. 8 L. 14 as:

The MTs are found to gradually adsorb at the interface from a droplet phase confirmed via the density of fluorescent label (supplementary Fig S1).

We have carefully checked the fluorescence intensity of MTs close to surface, and within a droplet. The results are newly given in Supplementary Fig.S1 (See the attached figures given below).

As seen in the figure, the MTs density decreases in the droplet while those close to the interface increased. As mentioned in the manuscript, adsorbed MTs then create cap-like structure. As long as this flux of the MTs were maintained, the convective flow in the droplet was observed.

The enthusiasm for the manuscript is somewhat diminished as it seems that the proposed mechanism is transient and the droplets on long time scales become quiescent.

Response:

To emphasize this point clearly, we replaced on P. 9 L. 15:

The lifetime of the convective motion may be extended by supplying kinesins, MTs and ATP to an active droplet from an external continuous phase.

As pointed, the convective motion of droplet may be recognized as transient dynamics while kinesin/MTs are adsorbed from bulk to the cap-like structure created on the droplet interface. We therefore mentioned that recharging of ATP and/or proteins are required to continuous motion of the droplet.

The authors claim that examining long-term dynamics is challenging due to the requirement to employ the ATP regeneration system. However, I would note that such a regeneration system has been employed in a recent complementary manuscript (ref 52), and this enabled the study of the dynamics over multiple hours.

Response:

The same question was raised from referee 2 in the first round. Introduction of ATP regeneration system change salt concentration and inevitably changes the phase separation behavior of PEG/DEX. Such change in the condition can alter the phase separation behavior and stability of micro droplet. We agree that such extension and prolonged life time of droplet motion is desirable, but we would like to leave it to the future study.

Replies to the comments by Referee 2

The authors have satisfactorily addressed nearly all reviewer questions in the revised and improved version of the manuscript. I would therefore recommend acceptance with minor revisions.

Two points related to my previous questions that the authors may want to consider:

1. Does the velocity of the droplet motion (stated to be 42.1 ± 7.4 nm/s on page 9) depend on kinesin density? The authors have addressed my question about varying motor activity through ATP concentration, which is apparently not the best knob to turn for the current setup. However, the active stress should depend on motor density as in the model Eq. 3, leading to stronger flows for higher C_K . It is not made clear if (or why not) this is observed experimentally. The authors should also be able to explore this easily with their modeling, but I could not find a reference to the droplet velocity measurement from simulations.

Response:

We have added these graphs in the supplementary figure S3, and have added the following sentence in the experimental results on P. 9 L. 9 as:

The velocity of the vortical flow shows positive correlation with the concentration of kinesin, C_K (Supplementary Fig. S3).

And in the numerical results on P. 13 L. 17 as:

The numerical simulations also reproduce the positive dependency of the flow velocity on C_K (Supplementary Fig. S3).

As shown in the manuscript the translational motion is an order of magnitude slower than the convective flow. We thus compared the speed of the flow as a measure of activity in the current system. Time course of the flow speed at the central part of the droplet, V_c , is shown below. Plotting averaged flow speed, V_c ($t = 1200-1770$ s) with respect to the kinesin density C_K , we find large V_c in the case of $C_K = 63$ nM. We found the flow speed increased further, when $C_K > 63$ nM, but could not follow by PIV analysis due to three dimensional nature of the convective rolls.

We also agree that the dependence of flow speed on parameter ζ , which corresponds to the kinesin density C_K , is indeed relevant information to compare our experiments with a numerical simulation. Numerical result reveals the increase of the maximum interfacial speed on C_K/C_{K0} , above the bifurcation point. To compare with the experiment, we also plotted the maximum flow velocity at the central part of the droplet. We find positive dependency of the flow velocity on parameter ζ . The reduced expression near the bifurcation point is also derived and agreed with the numerical simulation. These results show reasonable agreement with the experimental data with some deviation of quantitative aspect especially at low C_K region. As mentioned in the main text for the reason of the discrepancy observed in the phase diagram, the discrepancy could be due to the simplicity of our model, which overestimated the hydrostatic pressure at low C_M and/or underestimated the contractile stress at low C_K .

2. The authors now discuss the possible shape changes of active droplets as an important topic for future study. They conjecture that they do not observe such shape deformations in their system because: "We expect this to be explained by the contractile nature of our combination of MT and kinesin variants" Perhaps they can explain this a little more, because it is unclear to me why contractile stresses cannot cause deformations. At least in simulations, contractile stresses have been seen to produce droplet deformation, e.g. Ref. 56 Nickaen et al.

Response:

We replaced the sentence in the main text on P. 11 L. 8 as:

This could be explained by the manner of force generation by the adsorbed aster-like structure at the interface. Such structure may create stress mainly tangential, not normal to the interface. Revealing the mechanism underlying such effect represents a fascinating topic for future

study.

The difference from ref. 56 lies in the fact that the droplet has finite interfacial tension that constraints the deformation of the droplet, while ref. 56 neglect the membrane tension. However, we note that the contractile stress generated by the adsorbed kinesin/MT complex may deform droplet as seen by the bleb formation of cells. Thus, we realized that our comment “We expect this to be explained by the contractile nature of our combination of MT and kinesin variants” is inappropriate. We may instead suggest that the adsorbed structure of MT/kinesin create stress mainly tangential, not normal to the interface caused by the network structure affected by the presence of interface. We believe such force generation due to MT/kinesin complex adsorbed at the fluid-fluid interface is interesting open question for the future study.

Replies to the comments by Referee 3

The manuscript was largely approved and the authors responded satisfactorily to all my comments. One minor comment: the newly inserted sentence on the PIV analysis reads odd and should be revised for clarity. I support the publication of the manuscript and congratulate the authors to their sound work!

Response:

Thank you for the encouraging comment, and we realized the inserted sentence sound odd. We rephrased the sentence on P. 9 L. 1 as follows:

Based on Particle Image Velocimetry (PIV) analysis⁴⁶, the vortical flow inside of the droplet is visualized (Fig. 2d, schematically also depicted in Fig. 2b). We observed the flow from source to sink direction near the interface, and the strong backflow (sink to source direction) at the central part of the droplet.

REVIEWERS' COMMENTS:

Reviewer #1 (Remarks to the Author):

I believe it is fine to publish the manuscript in its present form.

Reviewer #2 (Remarks to the Author):

The authors have significantly improved the manuscript and satisfactorily answered my questions. In view of this and the fact that they demonstrate an important active matter phenomenon, spontaneously symmetry broken, phase separated, self-propelling cytoskeletal droplets, I recommend this manuscript for publication.